



# Relative effects of statistical preprocessing and postprocessing on a regional hydrological ensemble prediction system

Sanjib Sharma[1], Ridwan Siddique[2], Seann Reed[3], Peter Ahnert[3], Pablo Mendoza[4], Alfonso Mejia[1]

[1]Department of Civil and Environmental Engineering, The Pennsylvania State University, University Park, PA, USA
[2]Northeast Climate Science Center, University of Massachusetts, Amherst, MA, USA
[3]National Weather Service, Middle Atlantic River Forecast Center, State College, PA, USA
[4]Advanced Mining Technology Center (AMTC), Universidad de Chile, Santiago, Chile

*Correspondence to*: Alfonso Mejia (amejia@engr.psu.edu)

**Abstract.** The relative roles of statistical weather preprocessing and streamflow postprocessing in hydrological ensemble forecasting at short- to medium-range forecast lead times (day 1-7) are investigated. For this purpose, a regional hydrologic ensemble prediction system (RHEPS) is developed and implemented. The RHEPS is comprised by the following components: i) hydrometeorological observations (multisensor precipitation estimates, gridded surface temperature, and gauged streamflow); ii) weather ensemble forecasts (precipitation and near-surface temperature) from the National Centers for Environmental Prediction 11-member Global Ensemble Forecast System Reforecast version 2 (GEFSRv2); iii) NOAA's Hydrology Laboratory-Research Distributed Hydrologic Model (HL-RDHM); iv) heteroscedastic censored logistic regression (HCLR) as the statistical preprocessor; v) two statistical postprocessors, an autoregressive model with a single exogenous variable (ARX(1,1)) and quantile regression (QR); and vi) a comprehensive verification strategy. To implement the RHEPS, 1 to 7 days weather forecasts from the GEFSRv2 are used to force HL-RDHM and generate raw ensemble streamflow forecasts. Forecasting experiments are conducted in four nested basins in the U.S. middle Atlantic region, ranging in size from 381 to 12,362 km$^2$.

Results show that the HCLR preprocessed ensemble precipitation forecasts have greater skill than the raw forecasts. These improvements are more noticeable in the warm season at the longer lead times (>3 days). Both postprocessors, ARX(1,1) and QR, show gains in skill relative to the raw ensemble flood forecasts but QR outperforms ARX(1,1). Preprocessing alone has little effect on improving the skill of the ensemble flood forecasts. Indeed, postprocessing alone performs similar, in terms of the relative mean error, skill, and reliability, to the more involved scenario that includes both preprocessing and postprocessing. We conclude that statistical preprocessing may not always be a necessary component of the ensemble flood forecasting chain.

## 1 Introduction

The intersection of climate variability and change, increased exposure from expanding urbanization, and sea level rise are increasing the frequency of damaging flood events and making their prediction more challenging across the globe (Dankers et al., 2014; Wheater and Gober, 2015; Ward et al., 2015). Accordingly, current research and operational efforts in hydrological forecasting are seeking to develop and implement enhanced forecasting systems, with the goals of improving the skill and reliability of short- to medium-range flood forecasts (0-14 days), and providing more effective early warning services (Pagano et al., 2014; Thiemig et al., 2015; Emerton et al., 2016; Siddique and Mejia, 2017). Ensemble-based forecasting systems have become the preferred paradigm, showing substantial improvements over single-valued deterministic ones (Schaake et al., 2007; Cloke and Pappenberger, 2009; Demirel et al., 2013; Fan et al., 2014; Demargne et al., 2014; Schwanenberg et al., 2015; Siddique and Mejia, 2017). Ensemble flood forecasts can be generated in a number of ways, being the most common approach the use of meteorological forecast ensembles to force a hydrological model (Cloke and Pappenberger, 2009; Thiemig et al., 2015). Such meteorological forecasts can be generated by multiple alterations of a numerical weather prediction model, including perturbed initial conditions and/or multiple model physics and parameterizations.



A number of ensemble prediction systems (EPSs) are being used to generate flood forecasts. In the United States (U.S.), the NOAA's National Weather Service River Forecast Centers are implementing and using the Hydrological Ensemble Forecast Service to incorporate meteorological ensembles into their flood forecasting operations (Demargne et al., 2014; Brown et al., 2014). Likewise, the European Flood Awareness System from the European Commission (Alfieri et al., 2014) and the Flood Forecasting and Warming

Service from the Australia Bureau of Meteorology (Pagano et al., 2016) have adopted the ensemble paradigm. Furthermore, different regional EPSs have been designed and implemented for research purposes, meet specific regional needs, and/or real-time forecasting applications. Two examples, among several others (Zappa et al., 2008; Zappa et al., 2011; Hopson and Webster, 2010; Demuth and Rademacher, 2016; Addor et al., 2011; Golding et al., 2016; Bennett et al., 2014; Schellekens et al., 2011), are the Stevens Institute of Technology's Stevens Flood Advisory System for short-range flood forecasting (Saleh et al., 2016), and the National Center for

Atmospheric Research (NCAR)'s System for Hydromet Analysis, Research, and Prediction for medium-range streamflow forecasting (NCAR, 2017). Further efforts are underway to operationalize global ensemble flood forecasting and early warning systems, e.g., through the Global Flood Awareness System (Alfieri et al., 2013; Emerton et al., 2016).

EPSs are comprised by several system components. In this study, the Regional Hydrological Ensemble Prediction System (RHEPS) is used (Siddique and Mejia, 2017). The RHEPS is an ensemble-based research forecasting system, aimed primarily at

bridging the gap between hydrological forecasting research and operations by creating an adaptable and modular forecast emulator. The goal with the RHEPS is to facilitate the integration and rigorous verification of new system components, enhanced physical parameterizations, and novel assimilation strategies. For this study, the RHEPS is comprised by the following system components: i) precipitation and near surface temperature ensemble forecasts from the National Centers for Environmental Prediction 11-member Global Ensemble Forecast System Reforecast version 2 (GEFSRv2), ii) NOAA's Hydrology Laboratory-Research

Distributed Hydrologic Model (HL-RDHM) (Reed et al., 2004; Smith et al., 2012a; Smith et al., 2012b), iii) statistical weather preprocessor (hereafter referred to as preprocessing), iv) statistical streamflow postprocessor (hereafter referred to as postprocessing), v) hydrometeorological observations, and vi) verification strategy. Recently, Siddique and Mejia (2017) employed the RHEPS to produce and verify ensemble streamflow forecasts over some of the major river basins in the U.S. middle Atlantic region. Here, the RHEPS is specifically implemented to investigate the relative roles played by preprocessing and postprocessing

in enhancing the quality of ensemble flood forecasts.

The goal with statistical processing is to use statistical tools to quantify the uncertainty of and remove systematic biases in the weather and streamflow forecasts in order to improve the skill and reliability of forecasts. In weather and hydrological forecasting, a number of studies have demonstrated the benefits of separately implementing preprocessing (Sloughter et al., 2007; Verkade et al., 2013; Messner et al., 2014a; Yang et al., 2017) and postprocessing (Shi et al., 2008; Brown and Seo, 2010; Madadgar

et al., 2014; Ye et al., 2014; Wang et al., 2016; Siddique and Mejia, 2017). However, only a very limited number of studies have investigated the combined ability of preprocessing and postprocessing to improve the overall quality of ensemble streamflow forecasts (Kang et al., 2010; Zalachori et al., 2012; Roulin and Vannitsem, 2015). At first glance, in the context of medium-range streamflow forecasting, preprocessing seems necessary and beneficial since meteorological forcing are often biased and their uncertainty more dominant than the hydrological one (Cloke and Pappenberger, 2009; Bennett et al., 2014; Siddique and Mejia,

2017). In addition, some streamflow postprocessors assume unbiased forcing (Zhao et al., 2011) and hydrological models can be sensitive to forcing biases (Renard et al., 2010).

The few studies that have analyzed the joint effects of preprocessing and postprocessing on short- to medium-range streamflow forecasts have mostly relied on weather ensembles from the European Centre for Medium-range Weather Forecasts (ECMWF) (Zalachori et al., 2012; Roulin and Vannitsem, 2015; Benninga et al., 2016). Kang et al. (2010) used different forcing

but focused on monthly, as opposed to daily, streamflow. The conclusions from these studies have been mixed (Benninga et al.,



2016). Some have found statistical processing to be useful (Yuan and Wood, 2012), particularly postprocessing, while others have found that it contributes little to forecast quality. Overall, studies indicate that the relative effects of preprocessing and postprocessing depend strongly on the forecasting system (e.g., forcing, hydrological model, statistical processing technique, etc.) and conditions (e.g., lead time, study area, season, etc.), underscoring the research need to rigorously verify and benchmark new

forecasting systems that incorporate statistical processing.

The main objective of this study is to verify and assess the ability of preprocessing and postprocessing to improve ensemble flood forecasts from the RHEPS. This study differs from previous ones in several important respects. The assessment of statistical processing is done using a spatially distributed hydrological model whereas previous studies have tended to emphasize spatially lumped models. Much of the previous studies have used ECMWF forecasts, here we rely on GEFSRv2 precipitation and

temperature outputs. Also, we test and implement a preprocessor, namely heteroscedastic censored logistic regression (HCLR), which has not been used before in streamflow forecasting. We also consider a relatively wider range of nested, basin sizes and longer study period than in previous studies. In particular, this paper addresses the following questions:

- What are the separate and joint contributions of preprocessing and postprocessing over the raw RHEPS outputs?
- What forecast conditions (e.g., lead time, season, flood threshold, and basin size) benefit potential increases in skill?

- How much skill improvement can be expected from statistical processing under different uncertainty scenarios (i.e., when skill is measured relative to observed or simulated flow conditions)?

The remainder of the paper is organized as follows. Section 2 presents the study area. Section 3 describes the different components of the RHEPS. The main results and their implications are examined in section 4. Lastly, section 5 summarizes key findings.

**2 Study area**

The North Branch Susquehanna River basin in the U.S. middle Atlantic region (MAR) is selected as the study area (Fig. 1), with an overall drainage area of 12,362 km$^2$. The MAR is selected as flooding is an important regional concern. The MAR has a high level of urbanization and high frequency of extreme weather events, making it particularly vulnerable to damaging flood events (Gitro et al., 2014; MARFC, 2017). In the North Branch Susquehanna River basin, four different U.S. Geological Survey (USGS)

daily gauge stations, representing a system of nested subbasins, are selected as the forecast locations (Fig. 1). The selected locations are the Ostellic River at Cincinnatus (USGS gauge 01510000), Chenango River at Chenango Forks (USGS gauge 01512500), Susquehanna River at Conklin (USGS gauge 01503000), and Susquehanna River at Waverly (USGS gauge 01515000) (Fig. 1). The drainage area of the selected basins ranges from 381 to 12,362 km$^2$. Table 1 outlines some key characteristics of the study basins.

30                                   [Insert Figure 1 here]

[Insert Table 1 here]

**3 Approach**

In this section, we describe the different components of the RHEPS, including the hydrometeorological observations, weather forecasts, preprocessor, postprocessors, hydrological model, and the forecasting experiments and verification strategy.

**3.1 Hydrometeorological observations**



Three main observation datasets are used: multisensor precipitation estimates (MPEs), gridded near-surface air temperature, and daily streamflow. MPEs and gridded near-surface air temperature are used to run the hydrological model in simulation mode for parameter calibration purposes and to initialize the RHEPS. Both the MPEs and gridded near-surface air temperature data at 4 x 4 km$^2$ resolution were provided by the NOAA's Middle Atlantic River Forecast Center (MARFC). Similar to the NCEP stage-IV

dataset (Moore et al., 2015; Prat and Nelson, 2015), the MARFC's MPEs represent a continuous time series of hourly, gridded precipitation observations at 4 x 4 km$^2$ cells, which are produced by combining multiple radar estimates and rain gauge measurements. The gridded near-surface air temperature data at 4 x 4 km$^2$ resolution were developed by combining multiple temperature observation networks as described by Siddique and Mejia (2017). Daily streamflow observations for the selected basins were obtained from the USGS. The streamflow observations are used to verify the simulated flows, and the raw and

postprocessed ensemble streamflow forecasts.

### 3.2 Meteorological forecasts

GEFSRv2 data are used for the ensemble precipitation and near-surface air temperature forecasts. The GEFSRv2 uses the same atmospheric model and initial conditions as the version 9.0.1 of the Global Ensemble Forecast System and runs at T254L42 (~0.50º Gaussian grid spacing or ~55 km) and T190L42 (~0.67º Gaussian grid spacing or ~73 km) resolutions for the first and second 8

days, respectively (Hamill et al., 2013). The reforecasts are initiated once daily at 00 Coordinated Universal Time. Each forecast cycle consists of 3 hourly accumulations for day 1 to day 3 and 6 hourly accumulations for day 4 to day 16. In this study, we use 9 years of GEFSRv2 data, from 2004 to 2012, and forecast lead times from 1 to 7 days. The period 2004 to 2012 is selected to take advantage of data that were previously available to us (i.e., GEFSRv2 and MPEs for the MAR) from a recent verification study (Siddique et al., 2015). Forecast lead times of up to 7 days are chosen since we previously found that the GEFSRv2 skill is low

after 7 days (Siddique et al., 2015; Sharma et al., 2017).

### 3.3 Distributed hydrological model

NOAA's HL-RDHM is used as the spatially distributed hydrological model (Koren et al., 2004). Within HL-RDHM, the Sacramento Soil Moisture Accounting model with Heat Transfer (SAC-HT) is used to represent hillslope runoff generation, and the SNOW-17 module is used to represent snow accumulation and melting.

HL-RDHM is a spatially distributed conceptual model, where the basin system is divided into regularly spaced, square grid cells to account for spatial heterogeneity. Each grid cell acts as a hillslope capable of generating surface, interflow and groundwater runoff that discharges directly into the streams. The cells are connected to each other through the stream network system. Further, the SNOW-17 module allows each cell to accumulate snow and generate hillslope snow melt based on the near-surface air temperature. The hillslope runoff, generated at each grid cell by SAC-HT and SNOW-17, is routed to the stream network using a

nonlinear kinematic wave algorithm (Koren et al., 2004; Smith et al., 2012a). Likewise, flows in the stream network are routed downstream using a nonlinear kinematic wave algorithm that accounts for parameterized stream cross-section shapes ( Koren et al., 2004; Smith et al., 2012a). In this study, we run HL-RDHM using a 2-km horizontal resolution. Further information about the HL-RDHM can be found elsewhere (Koren et al., 2004; Reed et al., 2007; Smith et al., 2012a; Fares et al., 2014; Rafieeinasab et al., 2015; Thorstensen et al., 2016; Siddique and Mejia 2017).

To calibrate HL-RHDM, we first run the model using a-priori parameter estimates previously derived from available datasets (Koren et al., 2000; Reed et al., 2004; Anderson et al., 2006). We then select 10 out of the 17 SAC-HT parameters for calibration based upon prior experience and preliminary sensitivity tests. During the calibration process, each a-priori parameter field is multiplied by a factor. Therefore, we calibrate these factors instead of the parameter values at all grid cells, assuming that the a-





priori parameter distribution is true (e.g., Mendoza et al., 2012).The multiplying factors are adjusted manually first; once the manual changes do not yield noticeable improvements in model performance, the factors are tuned-up using stepwise line search (SLS; Kuzmin et al., 2008; Kuzmin, 2009). This method is readily available within HL-RDHM, and has been shown to provide reliable parameter estimates (Kuzmin et al., 2008; Kuzmin, 2009). With SLS, the following objective function is optimized:

$$OF = \sqrt{\sum_{i=1}^{m}[q_i - s_i(\Omega)]^2} , \qquad (1)$$

where $q_i$ and $s_i$ denote the daily observed and simulated flows at time $i$, respectively; $\Omega$ is the parameter vector being estimated; and $m$ is the total number of days used for calibration. Three years (2003-2005) of streamflow data are used to calibrate the HL-RDHM for the selected basins. The first year (year 2003) is used to warm-up HL-RDHM. To assess the model performance during calibration, we use the percent bias (PB), modified correlation coefficient ($R_m$), and Nash-Sutcliffe efficiency (NSE) (see appendix

for details). Note that these metrics are used during the manual phase of the calibration process, and to assess the final results from the implementation of the SLS. However, the actual implementation of the SLS is based on the objective function in Eq. (1).

### 3.4 Statistical weather preprocessor

Heteroscedastic censored logistic regression (HCLR) (Messner et al., 2014a; Yang et al., 2017) is implemented to preprocess the ensemble precipitation forecasts from the GEFSRv2. HCLR is selected since it offers the advantage, over other regression-based

preprocessors (Wilks, 2009), of obtaining the full, continuous predictive probability density function (pdf) of precipitation forecasts (Messner et al., 2014b). Also, HCLR has been shown to outperform other widely used preprocessors (Yang et al., 2017). In principle, HCLR fits the conditional logistic probability distribution function to the transformed (here the square root) ensemble mean and bias corrected precipitation ensembles. Note that we tried different transformations (square root, cube root, and fourth root), and found a similar performance between the square and cube root, both outperforming the fourth root. In addition, HCLR

uses the ensemble spread as a predictor, which allows the use of uncertainty information contained in the ensembles.

The development of the HCLR follows the logistic regression model initially proposed by Hamill et al. (2004) as well as the extended version of that model proposed by Wilks (2009). The extended logistic regression of Wilks (2009) is used to model the probability of binary responses such that

$$P(y \le z|x) = \Lambda[\omega(z) - \delta(x)], \qquad (2)$$

where $\Lambda(.)$ denotes the cumulative distribution function of the standard logistic distribution, $y$ is the transformed precipitation, $z$ is a specified threshold, $x$ is a predictor variable that depends on the forecast members, $\delta(x)$ is a linear function of the predictor variable $x$ , and the transformation $\omega(.)$ is a monotone nondecreasing function. Messner et al. (2014a) proposed the heteroscedastic extended logistic regression (HELR) preprocessor with an additional predictor variable $\varphi$ to control the dispersion of the logistic predictive distribution,

$$P(y \le z|x) = \Lambda\left\{\frac{\omega(z)-\delta(x)}{exp[\eta(\varphi)]}\right\}, \qquad (3)$$

where $\eta(.)$ is a linear function of $\varphi$. The functions $\delta(.)$ and $\eta(.)$ are defined as:

$$\delta(x) = a_0 + a_1 x, \text{ and} \qquad (4)$$

$$\eta(\varphi) = b_0 + b_1\varphi, \qquad (5)$$

where $a_0$, $a_1$, $b_0$, and $b_1$ are parameters that need to be estimated; $x = \frac{1}{K}\sum_{k=1}^{K} f_k^{\frac{1}{2}}$, i.e., the predictor variable $x$ is the mean of the



transformed, via the square root, ensemble forecasts $f$; $K$ is the total number of ensemble members; and $\varphi$ is the standard deviation of the square root transformed, precipitation ensemble forecasts.

Maximum likelihood estimation with the log-likelihood function is used to estimate the parameters associated with Eq. (3) (Messner et al., 2014a; Messner et al., 2014b). For this, the predicted probability $\pi_i$ of the $i^{\text{th}}$ observed outcome is determined. One

variation of the HELR preprocessor that can easily accommodate nonnegative variables, such as precipitation amounts, is HCLR, where $\pi_i$ is defined as (Messner et al., 2014b)

$$\pi_i = \begin{cases} \Lambda\left[\frac{\omega(0)-\delta(x)}{exp[\eta(\varphi)]}\right] & y_i = 0 \\ \lambda\left[\frac{\omega(y_i)-\delta(x)}{exp[\eta(\varphi)]}\right] & y_i > 0, \end{cases} \tag{6}$$

where $\lambda[.]$ denotes the likelihood function of the standard logistic function. As indicated by Eq. (6), HCLR fits a logistic error distribution with point mass at zero to the transformed predictand.

HCLR is applied here to each GEFSRv2 grid cell within the selected basins. At each cell, HCLR is implemented for the period 2004-2012 using a leave-one-out approach. For this, we select 7 years for training and the two remaining years for verification purposes. This is repeated until all the 9 years have been preprocessed and verified independently of the training period. This is done so that no training data is discarded and the entire 9-year period of analysis can be used to generate the precipitation forecasts. HCLR is employed for 6-hourly precipitation accumulations for lead times from 6 to 168 hours. To train the preprocessor,

we use a stationary training period, as opposed to a moving window, for each season and year to be forecasted, comprised by the seasonal data from all the 7 training years. Thus, to forecast a given season and specific lead time, we use ~6930 forecasts (i.e., 11 members x 90 days per season x 7 years). We previously tested using a moving window training approach and found that the results were similar to the stationary window one (Yang et al., 2017). To make the implementation of HCLR as straightforward as possible, the stationary window is used here. Finally, the Schaake Shuffle method as applied by Clark et al. (2004) is implemented

to maintain the observed space-time variability in the preprocessed GEFSRv2 precipitation forecasts. At each individual forecast time, the Schaake Shuffle is applied to produce a spatial and temporal rank structure for the ensemble precipitation values that is consistent with the ranks of the observations.

### 3.5 Statistical streamflow postprocessors

To statistically postprocess the flow forecasts generated by the RHEPS, two different approaches are tested, namely a first-order

autoregressive model with a single exogenous variable, ARX(1,1), and quantile regression (QR). We select the ARX(1,1) postprocessor since it has been suggested and implemented for operational applications in the U.S. (Regonda et al., 2013). QR is chosen because it is of similar complexity as the ARX(1,1) postprocessor but for some forecasting conditions it has been shown to outperform it (Mendoza et al., 2016). Furthermore, the ARX (1,1) and QR postprocessors have not been compared against each other for the forecasting conditions specified by the RHEPS. The postprocessors are implemented for the years 2004-2012, using

the same leave-one-out approach used for the preprocessor. The postprocessors are applied at each individual lead time from day 1 to 7. For this, the 6-hourly streamflow forecasts from HL-RDHM are averaged over 24 hours to get the streamflow forecast for a particular day.

### 3.5.1 First-order autoregressive model with a single exogenous variable

To implement the ARX(1,1) postprocessor, the observation and forecast data are first transformed into standard normal deviates

using the normal quantile transformation (NQT) (Krzysztofowicz, 1997; Bogner et al., 2012). The transformed observations and



forecasts are then used as predictors in the ARX(1,1) model (Siddique and Mejia, 2017). Specifically, for each forecast lead time, the ARX (1,1) postprocessor is formulated as follows:

$$q_{i+1}^T = (1 - c_{i+1})q_i^T + c_{i+1}f_{i+1}^T + \xi_{I+1},$$ (7)

where $q_i^T$ and $q_{i+1}^T$ are the NQT transformed observed flows at time steps $i$ and $i+1$, respectively; $c$ is the regression coefficient; $f_{i+1}^T$ is the NQT transformed forecast flow at time step $i+1$; and $\xi$ is the residual error term. In Eq. (7), assuming that there is significant correlation between $\xi_{i+1}$ and $q_i^T$, $\xi_{i+1}$ can be calculated as:

$$\xi_{i+1} = \frac{\sigma_{\xi_{i+1}}}{\sigma_{\xi_i}}\rho(\xi_{i+1},\xi_i)\xi_i + \vartheta_{i+1},$$ (8)

where $\sigma_{\xi_i}$ and $\sigma_{\xi_{i+1}}$ are the standard deviation of $\xi_i$ and $\xi_{i+1}$, respectively; $\rho(\xi_{i+1},\xi_i)$ is the serial correlation between $\xi_{i+1}$ and $\xi_i$; and $\vartheta_{i+1}$ is a random Gaussian error generated from $N(0,\sigma_{\vartheta_{i+1}}^2)$. To estimate $N(0,\sigma_{\vartheta_{i+1}}^2)$, the following equation is used:

$$\sigma_{\vartheta_{i+1}}^2 = [1 - \rho^2(\xi_{i+1},\xi_i)]\sigma_{\xi_{i+1}}^2.$$ (9)

To implement Eq. (7), ten equally spaced values of $c_{i+1}$ are selected from 0.1 to 0.9. For each value of $c_{i+1}$, $\sigma_{\vartheta_{i+1}}^2$ is determined from Eq. (9) using the training data to determine the other variables in Eq. (9). Then, $\vartheta_{i+1}$ is generated from $N(0,\sigma_{\vartheta_{i+1}}^2)$ and $\xi_{i+1}$ is calculated from Eq. (8). The result from Eq. (8) is used with Eq. (7) to generate a trace of $q_{i+1}^T$ which is transformed back to real space using the inverse NQT. These steps are repeated to generate multiple traces for each value of $c_{i+1}$. Lastly, the value of $c_{i+1}$ that produces the ensemble forecast with the smallest mean continuous ranked probability skill (CRPS) is selected. The ARX (1,1) postprocessor is applied at each individual lead time. For lead times beyond the initial one (day 1), one day-ahead predictions are used as the observed streamflow. For the cases where $q_{i+1}^T$ falls beyond the historical maxima, extrapolation is used by modeling the upper tail of the forecast distribution as hyperbolic (Journel and Huijbregts, 1978).

### 3.5.2 Quantile regression

Quantile regression (QR; Koenker and Bassett Jr, 1978; Koenker, 2005) is employed to determine the error distribution, conditional on the ensemble mean, resulting from the difference between observations and forecasts (Dogulu et al., 2015; López et al., 2014; Weerts et al., 2011; Mendoza et al., 2016). QR is applied here in streamflow space, since it has been shown that, in hydrological forecasting applications, QR has similar skill performance in streamflow and normal space (López et al., 2014). Another advantage of QR is that it does not make any prior assumptions regarding the shape of the distribution. Further, since QR results in conditional quantiles rather than conditional means, QR is less sensitive to the tail behavior of the streamflow dataset, and consequently, less sensitive to outliers. Note that although QR is here implemented separately for each lead time, the mathematical notation does not reflect this for simplicity.

The QR model is given by

$$\varepsilon_\tau' = d_\tau + e_\tau \bar{f},$$ (10)

where $\varepsilon_\tau'$ is the error estimate at quantile interval $\tau$; $\bar{f}$ is the ensemble mean; and $d_\tau$ and $e_\tau$ are the linear regression coefficients at $\tau$. The coefficients are determined by minimizing the sum of the residuals based on the training data as follows:

$$\min \sum_{i=1}^N w_\tau[\varepsilon_{\tau,i} - \varepsilon_\tau'(i,\bar{f}_i)],$$ (11)

$\varepsilon_{\tau,i}$ and $\bar{f}_i$ are the $i^{th}$ paired samples from a total of $N$ samples; $\varepsilon_{\tau,i}$ is computed as the observed flow minus the forecasted one, $q_\tau - f_\tau$; and $w_\tau$ is the weighting function for the $\tau^{th}$ quantile defined as:



$$w_\tau(\zeta_i) = \begin{cases} (\tau - 1)\zeta_i & if \ \zeta_i \le 0 \\ \tau\zeta_i & if \ \zeta_i > 0. \end{cases} \tag{12}$$

$\zeta_i$ is the residual term defined as the difference between $\varepsilon_{\tau,i}$ and $\varepsilon'_\tau(i, \bar{f}_i)$ for the quantile $\tau$. The minimization in Eq. (11) is solved using linear programming (Koenker, 2005).

Lastly, to obtain the calibrated forecast, $f_\tau$, the following equation is used:

$$f_\tau = \bar{f} + \varepsilon'_\tau. \tag{13}$$

In Eq. (13), the estimated error quantiles and the ensemble mean are added to form a calibrated discrete quantile relationship for a particular forecast lead time and thus generate an ensemble streamflow forecast.

### 3.6. Forecast experiments and verification

The verification analysis is carried out using the Ensemble Verification System (Brown et al., 2010). For the verification, the
following metrics are considered: relative mean error (RME), Brier skill score (BSS), mean continuous ranked probability skill score (CRPSS), and the decomposed components of the CRPS (Hersbach, 2000), i.e., the CRPS reliability (CRPS_rel) and CRPS potential (CRPS_pot). The definition of each of these metrics is provided in the appendix. Additional details about the verification metrics can be found elsewhere (Wilks, 2011;Jolliffe and Stephenson, 2012). Confidence intervals for the verification metrics are determined using the stationary block bootstrap technique (Politis and Romano, 1994), as done by Siddique et al. (2015). The
verification is focused on flood events by choosing flow amounts greater than that implied by a non-exceedance probability, in the sampled climatological probability distribution, of ~0.95. Thus, hereafter the term floods is used instead of streamflow to denote the forecasts generated by HL-RDHM. All the forecast verifications are done for lead times from 1 to 7 days.

To verify the forecasts for the period 2004-2012, six different forecasting scenarios are considered (Table 2). The first (S1) and second (S2) scenarios verify the raw and preprocessed ensemble precipitation forecasts, respectively. Scenarios 3 (S3), 4 (S4)
and 5 (S5) verify the raw, preprocessed, and postprocessed ensemble flood forecasts, respectively. The last scenario, S6, verifies the combined preprocessed and postprocessed ensemble flood forecasts. In S1 and S2, the raw and preprocessed ensemble precipitation forecasts are verified against the MPEs. For the verification of S1 and S2, each grid cell is treated as a separate verification unit. Thus, for a particular basin, the average performance is obtained by averaging the verification results from different verification units. The flood forecast scenarios, S3-S6, are verified against daily streamflow observations from the USGS.
The quality of the flood forecasts is evaluated conditionally upon forecast lead time, season (cool and warm), and flow threshold.

[Insert Table 2 here]

### 4 Results and discussion

This section is divided into four subsections. The first subsection demonstrates the performance of the spatially distributed model, HL-RDHM. The second subsection describes the performance of the raw and preprocessed GEFSRv2 ensemble precipitation
forecasts (forecasting scenarios S1 and S2). In the third subsection, the two statistical postprocessing techniques are compared. Lastly, the verification of different ensemble flood forecasting scenarios is shown in the fourth subsection (forecasting scenarios S3-S6).



### 4.1 Performance of the distributed hydrological model

To assess the performance of HL-RDHM, the model is used to generate streamflow simulations which are verified against daily observed flows, covering the entire period of analysis (years 2004-2012). Note that the simulated flows are obtained by forcing HL-RDHM with gridded precipitation and near surface temperature observations. The verification is done for the four basin outlets

shown in Fig. 1. To perform the verification and assess the quality of the streamflow simulations, the following statistical measures of performance are employed: modified correlation coefficient, $R_m$; Nash-Sutcliffe efficiency, NSE; and percent bias, PB. The mathematical definition of these metrics is provided in the appendix. The verification is done for both uncalibrated and calibrated simulation runs for the entire period of analysis. The main results from the verification of the streamflow simulations are summarized in Fig. 2.

[Insert Figure 2 here]

The performance of the calibrated simulation runs is satisfactory, with $R_m$ values ranging from ~0.75 to 0.85 (Fig. 2a). Likewise, the NSE, which is sensitive to both the correlation and bias, ranges from ~0.69 to 0.82 for the calibrated runs (Fig. 2b), while the PB ranges from ~5 to -11% (Fig. 2c). Relative to the uncalibrated runs, the $R_m$, NSE, and PB values improve by ~18, 29, and 47%, respectively. Further, the performance of the calibrated simulation runs is similar across the four selected basins,

although the largest size basin, WVYN6 (Fig. 2), seems to slightly outperform the other basins with $R_m$, NSE, and PB values of 0.85, 0.82, and -3% (Fig. 2), respectively. The lowest performance is seen in CNON6 with $R_m$, NSE, and PB values of 0.75, 0.7, and -11% (Fig. 2), respectively. Nonetheless, the performance metrics for both the uncalibrated and calibrated simulation runs do not deviate widely from each other in the selected basins, with perhaps the only exception being PB (Fig. 2c).

### 4.2 Verification of the raw and preprocessed ensemble precipitation forecasts

To examine the skill of both the raw and preprocessed GEFSRv2 ensemble precipitation forecasts, we plot in Fig. 3 the CRPSS (relative to sampled climatology) as a function of the forecast lead time (day 1 to 7) and season for the selected basins. Two seasons are considered: cool (October-March) and warm (April-September). Note that a CRPSS value of zero means no skill (i.e., same skill as the reference system) and a value of one indicates maximum skill. The CRPSS is computed using 6 hourly precipitation accumulations and high precipitation events. High precipitation events are here defined by an amount greater than that implied by

a non-exceedance probability, in the sampled climatological probability distribution, of ~0.95.

[Insert Figure 3 here]

The skill of both the raw and preprocessed ensemble precipitation forecasts tends to decline with increasing forecast lead time (Fig. 3). In the warm season (Figs. 3a-d), the CRPSS values vary overall, across all the basins, in the range from ~0 to 0.4 and from ~-0.2 to 0.3 for the preprocessed and raw forecasts, respectively; while in the cool season (Figs. 3e-h) the CRPSS values vary

overall in the range from ~0.1 to 0.6 and from 0 to 0.5 for the preprocessed and raw forecasts, respectively. The skill of the preprocessed ensemble precipitation forecasts tends to be greater than the raw ones across basins, seasons, and forecast lead times. Comparing the raw and preprocessed forecasts against each other, the relative skill gains from preprocessing are somewhat more apparent in the medium-range lead times (>3 days) and warm season. That is, the differences in skill seem not as significant in the short-range lead times (≤3 days). This seems particularly the case in the cool season where the confidence intervals for the raw and

preprocessed forecasts tend to overlap.

Indeed, seasonal skill variations are noticeable in all the basins. Even though the relative gain in skill from preprocessing is slightly greater in the warm season, the overall skill of both the raw and preprocessed forecasts is better in the cool season than the warm one. This may be due, among other potential factors, to the greater uncertainty associated with modeling convective precipitation, which is more prevalent in the warm season, by the NWP model used to generate the GEFSRv2 outputs (Hamill et



al., 2013; Baxter et al., 2014). Nonetheless, the warm season preprocessed forecasts show gains in skill across all the lead times and basins. For a particular season, the forecast ensembles across the different basins tend to display similar performance; i.e. the analysis does not reflect skill sensitivity to the basin size as in other studies (Siddique et al., 2015; Sharma et al., 2017). This is expected here since the verification is performed for each GEFSRv2 grid cell, rather than verifying the average for the entire basin.

That is, the results in Fig. 3 are for the average skill performance obtained from verifying each individual grid cell within the selected basins.

Based on the results presented in Fig. 3, we may expect some skill contribution to the flood ensembles from forcing the HL-RDHM with the preprocessed precipitation, as opposed to using the raw forecast forcing. Although the contribution may not be as large, since the differences between the preprocessed and raw precipitation forecasts are only mild. It may also be expected that

the contributions are greater for the medium-range lead times and warm season. This will be examined in subsection 4.4, prior to that we compare next the two postprocessors, namely ARX(1,1) and QR.

### 4.3 Selection of the flood postprocessor

The ability of the ARX(1,1) and QR postprocessors to improve ensemble flood forecasts is investigated here. The postprocessors are applied to the raw flood ensembles at each forecast lead time from day 1 to 7. To examine the skill of the postprocessed flood

forecasts, Fig. 4 displays the CRPSS (relative to the raw ensemble flood forecasts) versus the forecast lead time for all the selected basins, for both cool (Fig. 4a-d) and warm (Fig. 4e-h) seasons. The overall tendency is for both postprocessing techniques to demonstrate improved forecast skill across all the basins, seasons, and most of the lead times. The skill can improve as much as 40% at the later lead times (Fig. 4b). The general trend in Fig. 4 is for the skill of the postprocessors to increase with increasing lead time. Note that this is the case since the skill is here measured relative to the raw flood forecasts which is done to better isolate

the effect of the postprocessors on the flood forecasts. This means that the postprocessors are more able to improve the medium-range (>3 days) forecasts than the short-range (≤3 days) ones.

[Insert Figure 4 here]

The gains in skill from QR vary from ~5% (Fig. 4a at the day 1 lead time) to 40% (Fig. 4b at the day 5 lead time) depending upon the season and lead time. While the gains from ARX(1,1) vary from ~4% (Fig. 4e at the day 1 lead time) to a much lower

level of ~22% (Fig. 4c at the day 2 lead time). In most cases, both postprocessors exhibit somewhat similar performance at the initial lead times (days 1-2), with skills varying from nearly 0.1 (e.g., Figs. 4a and 4e) to 0.4 (Fig. 4f at the day 2 lead time). At the later lead times (4-7 days), QR tends to outperform ARX(1,1), with the difference in performance being as high as 30% (Fig. 4d at the day 7 lead time). This is noticeable across all the basins and for both seasons. The skill improvement of QR over ARX(1,1) is significant at later lead times (> day 3), as indicated by the fact that the confidence intervals for the curves representing the

postprocessors in Fig. 4 often do not overlap. There are also seasonal differences in the performance of the postprocessors. In particular, the gains in skill from ARX(1,1) in the warm season can be quite low (Figs. 4a and 4c).

As discussed and demonstrated in Fig. 4, QR performs better than ARX(1,1). Indeed, we also found (plots not shown) that QR displays better reliability than ARX(1,1) across lead times, basins, and seasons. Therefore, we select QR as the statistical flood postprocessor to examine the interplay between preprocessing and postprocessing in the RHEPS.

### 4.4 Verification of the ensemble flood forecasts for different statistical processing scenarios

In this subsection, we examine the effects of different statistical processing scenarios on the ensemble flood forecasts from the RHEPS. Recall that, to consider flood events, the verification is done for flow events with an amount greater than that implied by a non-exceedance probability, in the sampled climatological probability distribution, of ~0.95. The forecasting scenarios





considered here are S3-S6 (Table 1 defines the scenarios). To facilitate presenting the verification results, this subsection is divided into the following four parts: relative mean error, CRPSS, CRPS decomposition, and BSS.

### 4.4.1 Relative mean error

To examine the bias associated with the mean ensemble flood forecasts under scenarios S3-S6, we plot the RME versus the forecast
lead time for all the basins (Fig. 5), and the warm (Fig. 5a-d) and cool seasons (Fig. 5e-h). Results in Fig. 5 show that, under all the considered scenarios, the mean ensemble flood forecasts exhibit underforecasting bias across basins, lead times, and seasons. The underforecasting bias increases with the lead time, and decreases somewhat with the increase in basin size. For example, the bias for the largest basin, WVYN6, is -0.1 at the day 1 lead time and scenario S3 (Fig. 5d), while for the same lead time and scenario the bias is -0.35 for the smallest basin (Fig. 5a). In essence, the GEFSRv2-based flood ensembles exhibit a conditional
bias that is consistent with the conditional bias (i.e., to significantly underforecast large events) for the GEFSRv2 precipitation ensembles (Siddique et al., 2015; Sharma et al., 2017).

[Insert Figure 5 here]

The two most striking features of Fig. 5 are: i) the significant difference in performance between the pair S3-S4 and S5-S6 and, in contrast, ii) the similarity in performance between S5 and S6. The former confirms that statistical processing, in particular
postprocessing, has a significant effect on the flood ensembles. Recall that to generate the ensemble flood forecasts S5 only employs postprocessing, while S6 considers both preprocessing and postprocessing (Table 1). Yet, the RME across basins, lead times, and seasons for both S5 and S6 are quite similar, with differences tending to be not as significant. The similarity between S5 and S6 indicates that in this case preprocessing has a mild effect on the flood forecasts.

As a corollary to the latter comment, it can be argued that by only postprocessing the raw flood ensembles most of the benefits
from statistical processing can be realized. This seems also supported by the results for S3 and S4 (Fig. 5). The differences between the RME of the flood forecasts generated by forcing the HL-RDHM with raw, S3, versus preprocessed precipitation ensembles, S4, are only significant at lead times greater than 4 days. In addition, the differences are not as large, with the largest one being ~-0.18 at the day 5 lead time in CNON6 (Fig. 5b). This is not entirely surprising as we previously saw (Fig. 3) that differences between the raw and preprocessed precipitation ensembles are only significant at the later lead times where the skill of the forecast
is, in any case, already somewhat low. In terms of the seasonal analysis, both S5 and S6 tend to be less biased in the cool season than in the warm one, particularly at the short-range lead times (<3 days). This can be seen by comparing Fig. 4c against Fig. 4g at the day 1 lead time. The role played by preprocessing and postprocessing in ensemble flood forecasting is further evaluated next in terms of the forecast skill.

### 4.4.2 CRPSS

The skill of the ensemble flood forecasts for S3-S6 is assessed using the CRPSS relative to the sampled climatology (Fig. 6). Fig. 6 shows that, across lead times, basin sizes, and seasons, the results for the CRPPS are qualitatively similar to those for the RME (Fig. 5). That is, the most salient feature of Fig. 6 is that the performance of the flood forecasts tends to progressively improve from S3 to S6. This means that the forecast skill tends to improve across lead times, basin sizes, and seasons as additional statistical processing steps are included in the RHEPS' forecasting chain. The skill first increases from the raw scenario (i.e., S3 where no
statistical processing is done) to the scenario where only preprocessing is performed, S4. However, the gain in skill between S3 and S4 is generally small, particularly at the short lead times, reinforcing the fact that preprocessing may have little effect on the flood forecasts. The skill then shows a more significant improvement for both S5 and S6, relative to S4. As was the case with the



RME, the differences in skill between S5 and S6 are not as significant, suggesting that postprocessing alone (i.e., without preprocessing) may be sufficient to remove systematic biases in the flood forecasts.

[Insert Figure 6 here]

In terms of the warm and cool seasons, at the initial forecast lead times ($\leq 2$ days), the skill of the flood forecasts tends to be slightly greater in the cool season (Figs. 6e-h) than in the warm one (Figs. 6a-d), with the exception of CNON6. As was the case in the calibration results (e.g., in Fig. 2c), during the cool season CNON6 has a lower performance prior to postprocessing (S3 or S4 in Fig. 6f) than the other basins. Interestingly, after postprocessing (S5 in Fig. 6f), the skill of CNON6 is as good as that of CINN6, even though at the day 1 lead time the skill for S3 is ~0.3 for CNON6 (Fig. 6f) and ~0.5 for CINN6 (Fig. 6e). Hence, the postprocessor seems capable to compensate some for the lesser performance of CNON6 during calibration.

### 4.4.3 CRPS decomposition

Fig. 7 displays different components of the mean CRPS against lead times of 1, 3, and 7 days for all the basins according to both the warm (Figs. 7a-d) and cool (Figs. 7e-h) seasons. The components presented here are reliability ($CRPS_{rel}$) and potential CRPS ($CRPS_{pot}$) (Hersbach, 2000). $CRPS_{rel}$ measures the average reliability of the ensemble forecasts across all the possible events, i.e., it examines whether the fraction of observations that fall below the $j$-th of $n$ ranked ensemble members is equal to $j/n$ on average. $CRPS_{pot}$ represents the lowest possible CRPS that could be obtained if the forecasts were made perfectly reliable (i.e., $CRPS_{rel}=0$). Note that the CRPS, $CRPS_{rel}$, and $CRPS_{pot}$ are all negatively oriented, with perfect score of zero. Overall, as was the case with the RME (Fig. 5) and CRPSS (Fig. 6), the CRPS decomposition reveals that forecasts reliability increases from S3 to S6.

[Insert Figure 7 here]

Interestingly, improvements in forecast quality for S5 and S6, relative to the raw flood forecasts of S3, are mainly due to reductions in $CRPS_{rel}$ (i.e., by making the forecasts more reliable), whereas for S4 better forecast quality is achieved by reductions in $CRPS_{pot}$. The latter is seen across all basins, lead times, and seasons. The explanation for this lies in the implementation of the HCLR preprocessor, which uses the ensemble spread as a predictor of the dispersion of the predictive pdf and the $CRPS_{pot}$ is sensitive to the spread (Messner et al., 2014a). Although the forecasts from S3 have lower $CRPS_{pot}$, the forecasts including postprocessing, S5 and S6, ultimately result in lower CRPS. This indicates that the forecasts for S5 and S6 are more reliable than for S3 and S4.

### 4.4.4 BSS

In our final verification comparison, the BSS of the ensemble flood forecasts for S5 (Figs. 8a-d) and S6 (Figs. 8e-h) are plotted against the non-exceedance probability associated with different flood thresholds ranging from 0.95 to 0.99. The BSS is computed for all the basins, warm season, and lead times of 1, 3 and 7 days. In addition, the BSS is computed relative to both observed (solid lines in Fig. 8) and simulated (dashed lines in Fig. 8) floods. When the BSS is computed relative to observed floods, it considers the effect on forecast skill of both meteorological and hydrological uncertainties. While the BSS relative to simulated floods is mainly affected by meteorological uncertainties. The difference between the two, i.e., the BSS relative to observed floods minus the BSS relative to simulated ones, provides an estimate of the effect of hydrological uncertainties on the skill of the flood forecasts. Similar to the CRPSS, the BSS value of zero means no skill (i.e., same skill as the reference system) and a value of one indicates perfect skill.

[Insert Figure 8 here]

In general, the skill of flood forecasts tends to decrease with lead time across the flow thresholds and basins. As was the case with the CRPSS (Fig. 6), the BSS values appear similar for S5 (Figs. 8a-d) and S6 (Figs. 8e-h). The only exception is CKLN6




(Figs. 8c and 8g) where, at the higher flood thresholds, S6 has better skill than S5 at the day 1 and 3 lead times. With respect to the basin size, the skill tends to improve some from the small to the large basin. For instance, for non-exceedance probabilities of 0.95 and 0.99 at the day 1 lead time, the BSS values for the smallest basin (Fig. 8a), measured relative to the observed flows, are ~0.49 and 0.35, respectively. For the same conditions, both values increase to ~0.65 for the largest basin (Fig. 8d).

Indeed, the most notable feature in Fig. 8 is that the effect of hydrological uncertainties on forecast skill is evident at the day 1 lead time, while meteorological uncertainties clearly dominate at the day 7 lead time. With respect to the latter, notice that the solid and dashed green lines for the day 7 lead time tend to be very close to each other in Fig. 8, indicating that hydrological uncertainties are relatively small compared to meteorological ones. Hydrological uncertainties are largest at the day 1 lead time, particularly for the small basins (Figs. 8a-b and 8e-f). For example, for a non-exceedance probability of 0.95 and at a day 1 lead time (Fig. 8b), the BSS value relative to the simulated and observed floods are ~0.79 and 0.38, respectively, suggesting a reduction of ~50% skill due to hydrological uncertainties.

## 5 Summary and conclusion

In this study, we used the RHEPS to investigate the effect of statistical processing on short- to medium-range ensemble flood forecasts. First, we assessed the raw precipitation forecasts from the GEFSRv2 (S1), and compared them with the preprocessed precipitation ensembles (S2). Then, flood ensembles were generated with the RHEPS for four different forecasting scenarios involving no statistical processing (S3), preprocessing alone (S4), postprocessing alone (S5), and both preprocessing and postprocessing (S6). The verification of ensemble precipitation and flood forecasts was done for the years 2004-2012, using four basins in the U.S. MAR. We found that – for the models, datasets, and study domain used here - the skill gains from joint preprocessing and postprocessing are similar to those from postprocessing alone. Other specific findings are as follows:

- The HCLR preprocessed ensemble precipitation forecasts show improved skill relative to the raw forecasts. The improvements are more noticeable in the warm season at the longer lead times (>3 days).

- Both postprocessors, ARX(1,1) and QR, show gains in skill relative to the raw ensemble flood forecasts. For the medium-range lead times (>3 days), the gains with QR, however, tend to be greater than with ARX(1,1), particularly during the warm season.

- By comparing different statistical processing scenarios for the ensemble flood forecasts, it was found that the scenario with preprocessing alone has little effect on improving the skill of the flood forecasts in contrast with the postprocessing alone scenario.

- The scenario including only postprocessing performs similar, in terms of the relative mean error, CRPSS, and reliability, to the more complex scenario consisting of both preprocessing and postprocessing. It thus seems for our conditions, using GEFSRv2 forecasts, that preprocessing may be unnecessary.

- The skill of the postprocessing alone scenario and the scenario that combines preprocessing and postprocessing was further assessed using the Brier skill score for different flood thresholds. This assessment further confirmed that both scenarios have similar skill and performance behavior.

These conclusions are specific to the RHEPS forecasting system, which is mostly relevant to the U.S. research and operational communities as it relies on a weather and a hydrological model that are used in this domain. However, the use of a global weather forecasting system illustrates the potential of applying the statistical techniques tested here in other regions worldwide.

The emphasis of this study has been on benchmarking the contributions of statistical processing to the RHEPS. To accomplish this, our approach required that the quality of ensemble flood forecasts be verified over multiple years (i.e., across many flood

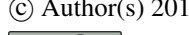



cases) to obtain robust verification statistics. Future research, however, could be focused on studying how distinct hydrological processes contribute or constrain forecast quality. This effort could be centered around specific flood events rather than in the statistical, many-cases approach taken here. To further assess the relative importance of the various components of the RHEPS, additional tests involving the uncertainty to initial hydrologic conditions and hydrological parameters could be performed. For

instance, the combined use of data assimilation and postprocessing has been shown to produce more reliable and sharper streamflow forecasts (Bourgin et al., 2014). The potential for the interaction of preprocessing and postprocessing with data assimilation to significantly enhance streamflow predictions, however, has not been investigated. This could be investigated in the future with the RHEPS, as the pairing of data assimilation with preprocessing and postprocessing could facilitate translating the improvements in the preprocessed meteorological forcing down the hydrological forecasting chain.

*Data availability*: Daily streamflow observation data for the selected forecast stations can be obtained from the USGS website (https://waterdata.usgs.gov/nwis/). Multisensor precipitation estimates are obtained from the NOAA's Middle Atlantic River Forecast Center. Precipitation and temperature forecast datasets can be obtained from the NOAA Earth System Research Laboratory website (https://www.esrl.noaa.gov/psd/forecasts/reforecast2/download.html).

**Appendix A: Verification metrics**

**Modified correlation coefficient ($R_m$):** The modified version of the correlation coefficient, called as modified correlation coefficient $R_m$, compare event specific observed and simulated hydrographs (McCuen and Snyder, 1975). In the modified version, an adjustment factor based on the ratio of the observed and simulated flow is introduced to refine the conventional correlation coefficient $R$. The modified correlation coefficient $R_m$ is defined as:

$$R_m = R \frac{min\{\sigma_s, \sigma_q\}}{max\{\sigma_s, \sigma_q\}},$$  (A1)

where $\sigma_s$ and, $\sigma_q$ denote the standard deviation of the simulated and observed flows, respectively.

**Percent bias (PB):** PB measures the average tendency of the simulated flows to be larger or smaller than their observed counterparts. Its optimal value is 0.0 where positive values indicate model overestimation bias, and negative values indicate model

underestimation bias. The PB is estimated as follows:

$$PB = \frac{\sum_{i=1}^{N}(s_i - q_i)}{\sum_{i=1}^{N} q_i} \times 100,$$  (A2)

where $s_i$ and $q_i$ denote the simulated and observed flow, respectively, at time $i$.

**Nash-Sutcliffe efficiency (NSE):** The NSE (Nash and Sutcliffe, 1970) is defined as the ratio of the residual variance to the initial

variance. It is widely used to indicate how well the simulated flows fit the observations. The range of NSE can vary between negative infinity to 1.0, with 1.0 representing the optimal value and values should be larger than 0.0 to indicate minimally acceptable performance. The NSE is computed as follows:

$$NSE = 1 - \frac{\sum_{i=1}^{N}(s_i - q_i)^2}{\sum_{i=1}^{N}(q_i - \bar{q}_i)^2},$$  (A3)

where $s_i$, $q_i$, and $\bar{q}_i$ are the simulated, observed, and mean observed flow, respectively, at time $i$ .



**Relative mean error (RME):** RME quantifies the average error between the ensemble mean forecast and their corresponding observation as a fraction of the averaged observed value. RME gives an indication how good the forecast is relative to the observation. RME is expressed as follows:

$$RME = \frac{\sum_{i=1}^{n}(\bar{f}_i - q_i)}{\sum_{i=1}^{N} q_i},$$
(A4)

where $\bar{f}_i = \frac{1}{m}\sum_{k=1}^{m} f_{i,k}$, $m$ is the number of ensemble members, $f_{i,k}$ is the forecast for member $k$ and time $i$, $q_i$ denotes the corresponding observation at time $i$, and $n$ denotes the total number of pairs of forecasts and observed values.

**Brier Skill Score (BSS):** The Brier score (BS; Brier, 1950) is analogous to the mean squared error, but where the forecast is a

probability and the observation is either a 0.0 or 1.0. The BS is given by

$$BS = \frac{1}{n}\sum_{i=1}^{n}\left[F_{f_i}(z) - F_{q_i}(z)\right]^2,$$
(A5)

where the probability of $f_i$ to exceed a fixed threshold $z$ is

$$F_{f_i}(z) = P_r[f_i > z],$$
(A6)

$n$ is again the total number of forecast-observation pairs, and

$$F_{q_i}(z) = \begin{cases} 1, & q_i > z \\ 0, & otherwise. \end{cases}$$
(A7)

In order to compare the skill score of the main forecast system with respect to the reference forecast, it is convenient to define the Brier Skill Score (BSS):

$$BSS = 1 - \frac{BS_{main}}{BS_{reference}},$$
(A8)

where $BS_{main}$ and $BS_{reference}$ are the BS values for the main forecast system (i.e. the system to be evaluated) and reference

forecast system, respectively. Any positive values of the BSS, from 0 to 1, indicate that the main forecast system performs better than the reference forecast system. Thus, a BSS of 0 indicates no skill and a BSS of 1 indicates perfect skill.

**Mean Continuous Ranked Probability Skill Score (CRPSS):** Continuous Ranked Probability Score (CRPS) quantifies the integrated square difference between the cumulative distribution function (cdf) of a forecast, $F_f(z)$, and the corresponding cdf of

the observation, $F_q(z)$. The CRPS is given by

$$CRPS = \int_{-\infty}^{\infty}\left[F_f(z) - F_q(z)\right]^2 dz.$$
(A9)

To evaluate the skill of the main forecast system relative to the reference forecast system, the associated skill score, the mean Continuous Ranked Probability Skill Score (CRPSS), is defined as:

$$CRPSS = 1 - \frac{CRPS_{main}}{CRPS_{reference}},$$
(A10)

where the CRPS is averaged across $n$ pairs of forecasts and observations to calculate the mean CRPS of the main forecast system ($CRPS_{main}$) and reference forecast system ($CRPS_{reference}$). The CRPSS varies from -∞ to 1. Any positive values of the CRPSS, from 0 to 1, indicate that the main forecast system performs better than the reference forecast system.





To further explore the forecast skill, the $CRPS_{main}$ is decomposed into the CRPS reliability ($CRPS_{rel}$) and potential($CRPS_{pot}$) such that Hersbach (2000)

$$CRPS_{main} = CRPS_{rel} + CRPS_{pot}. \tag{A11}$$

The $CRPS_{rel}$ measures the average reliability of the precipitation ensembles similarly to the rank histogram, which shows whether

the frequency that the verifying analysis was found in a given bin is equal for all bins (Hersbach 2000). The $CRPS_{pot}$ measures the CRPS that one would obtain for a perfect reliable system. It is sensitive to the average ensemble spread and outliers.

*Acknowledgements:* We acknowledge the funding support provided by the NOAA/NWS through Award NA14NWS4680012.

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



**Table 1**. Main characteristics of the four study basins.

| Location of outlet | Cincinnatus, New York | Chenango Forks, New York | Conklin, New York | Waverly, New York |
|---|---|---|---|---|
| NWS id | CINN6 | CNON6 | CKLN6 | WVYN6 |
| USGS id | 01510000 | 01512500 | 01503000 | 01515000 |
| Area [$km^2$] | 381 | 3841 | 5781 | 12362 |
| Latitude | $42^032'28''$ | $42^013'05''$ | $42^002'07''$ | $41^059'05''$ |
| Longitude | $75^053'59''$ | $75^050'54''$ | $75^048'11''$ | $76^030'04''$ |
| Minimum daily flow[*] [$m^3/s$] | 0.31 (0.11) | 4.05 (2.49) | 6.80 (5.32) | 13.08 (6.71) |
| Maximum daily flow[*] [$m^3/s$] | 172.73 (273.54) | 1248.77 (1401.68) | 2041.64 (2174.734) | 4417.42 (4417.42) |
| Mean daily flow[*] [$m^3/s$] | 8.89 (9.17) | 82.36 (81.66) | 122.93 (121.99) | 277.35 (215.01) |
| Climatological flow (Pr=0.95)[**] [$m^3/s$] | 29.45 | 266.18 | 382.28 | 843.84 |

[*]The number in parenthesis is the historical (based on entire available record, as opposed to the period 2004-2012 used in this study) daily minimum, maximum, or mean recorded flow.
[**]Pr=0.95 indicates flows with exceedance probability of 0.05.





**Table 2**. Summary and description of the verification scenarios.

| Scenario | Description |
|---|---|
| S1 | Verification of the raw ensemble precipitation forecasts from the GEFSRv2 |
| S2 | Verification of the preprocessed ensemble precipitation forecasts from the GEFSRv2: GEFSRv2+HCLR |
| S3 | Verification of the raw ensemble flood forecasts: GEFSRv2+HL-RDHM |
| S4 | Verification of the preprocessed ensemble flood forecasts: GEFSRv2+HCLR+HL-RDHM |
| S5 | Verification of the postprocessed ensemble flood forecasts: GEFSRv2+HL-RDHM+QR |
| S6 | Verification of the preprocessed and postprocessed ensemble flood forecasts: GEFSRv2+HCLR+HL-RDHM+QR |



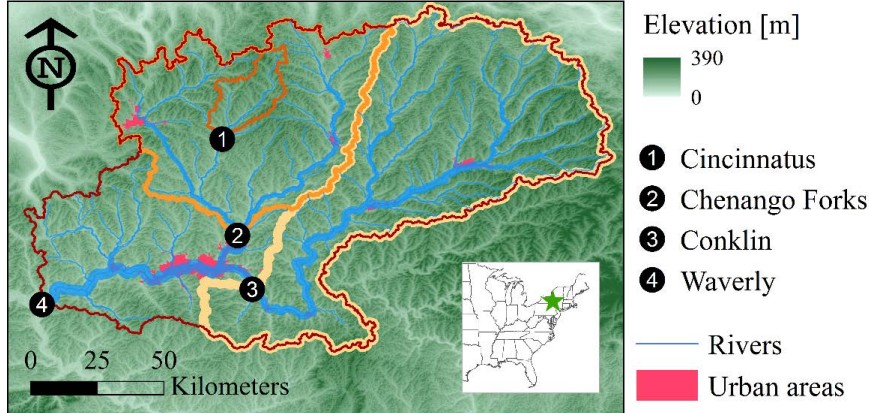

Figure 1: Map illustrating the location of the four selected river basins in the U.S. middle Atlantic region.





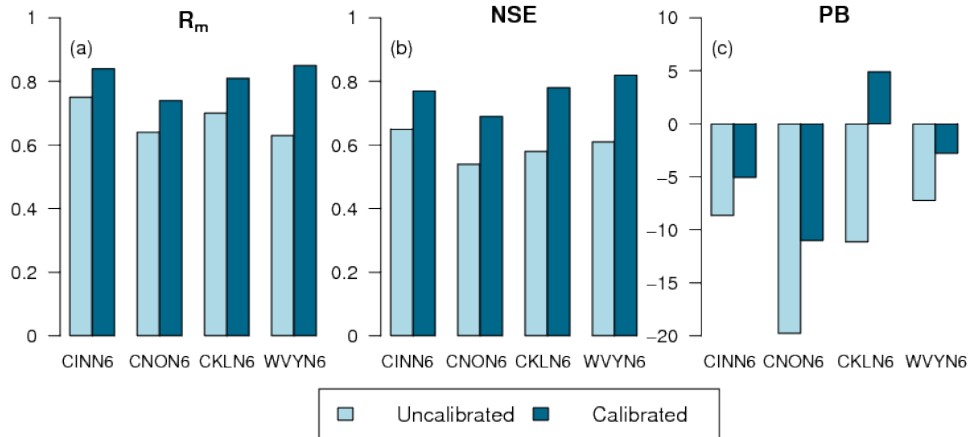

**Figure 2: Performance statistics for the uncalibrated and calibrated simulation runs for the entire period of analysis (years 2004-2012):**
**(a) $R_m$, (b) NSE, and (c) PB.**

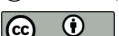



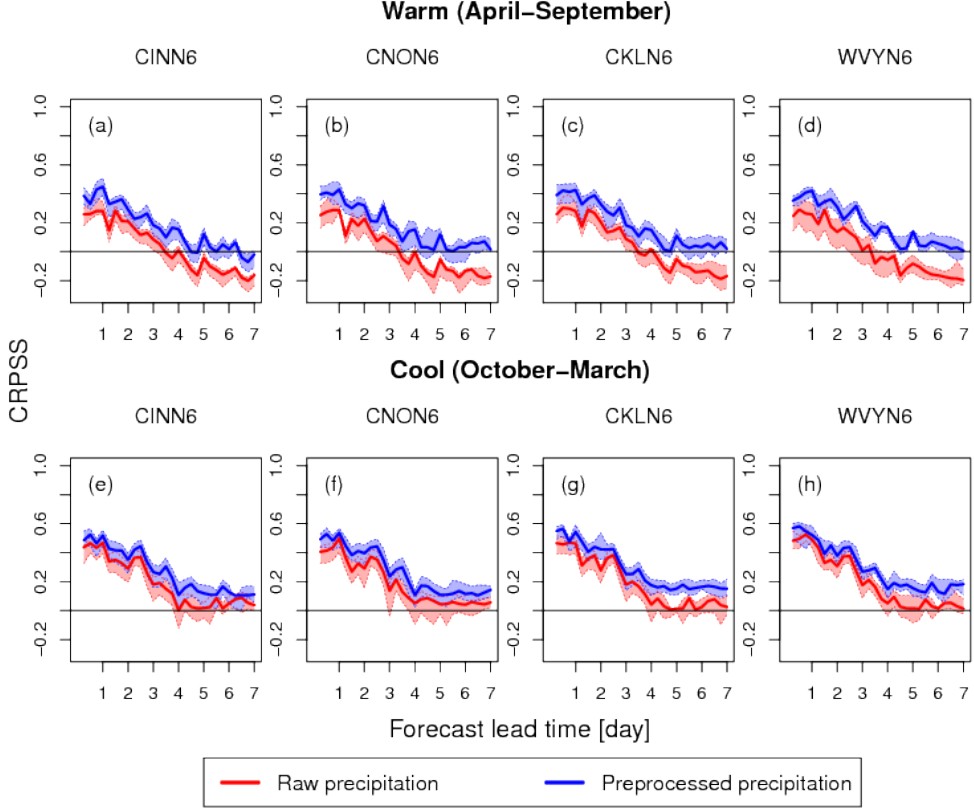

**Figure 3: CRPSS (relative to sampled climatology) of the raw (red curves) and preprocessed (blue curves) ensemble precipitation forecasts from the GEFSRv2 vs the forecast lead time during the (a)-(d) warm (April-September) and (e)-(h) cool season (October-March) for the selected basins.**





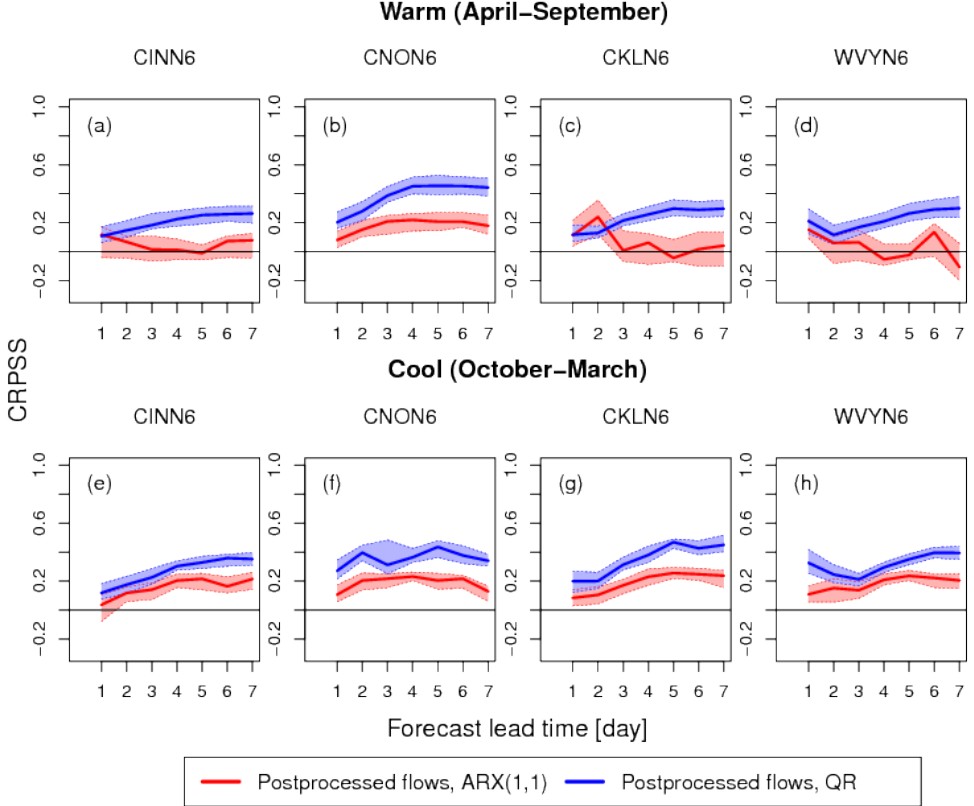

**Figure 4: CRPSS (relative to the raw forecasts) of the ARX(1,1) (red curves) and QR (blue curves) postprocessed ensemble flood forecasts vs the forecast lead time during the (a)-(d) warm (April-September) and (e)-(h) cool season (October-March) for the selected basins.**



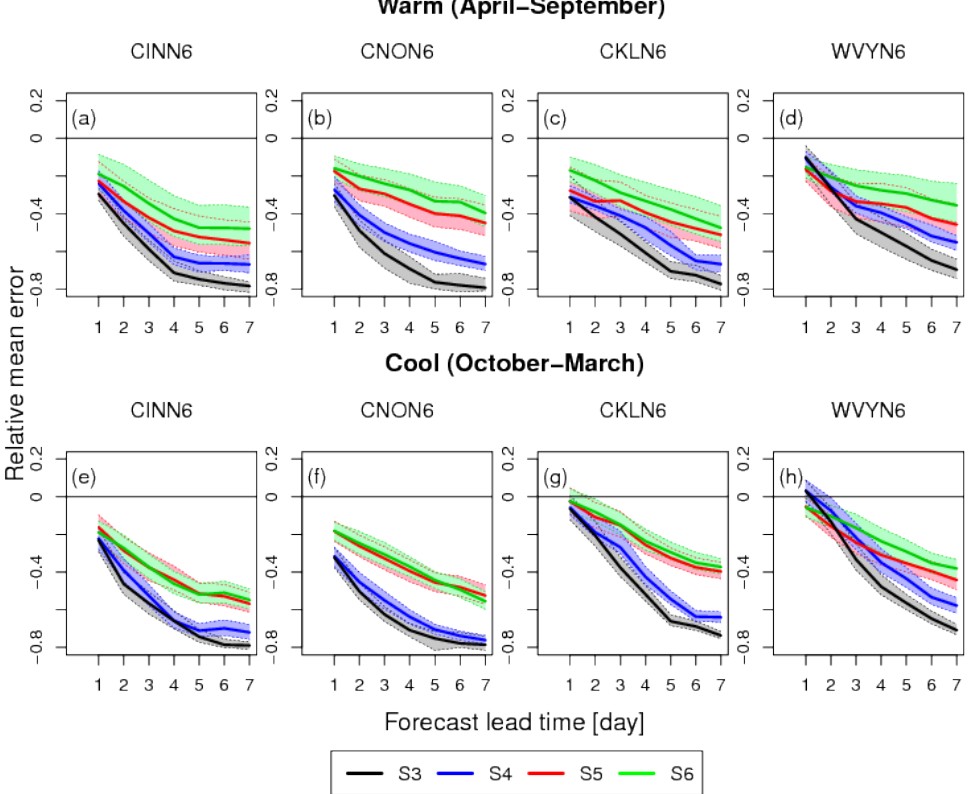

Figure 5: Relative mean error (RME) of the mean ensemble flood forecasts vs the forecast lead time during the (a)-(d) warm (April-September) and (e)-(h) cool season (October-March) for the selected basins. The curves represent the different forecasting scenarios S3-S6. Note that S3 consists of GEFSRv2+HL-RDHM, S4 of GEFSRv2+HCLR+HL-RDHM, S5 of GEFSRv2+HL-RDHM+QR, and S6 of GEFSRv2+HCLR+HL-RDHM+QR.



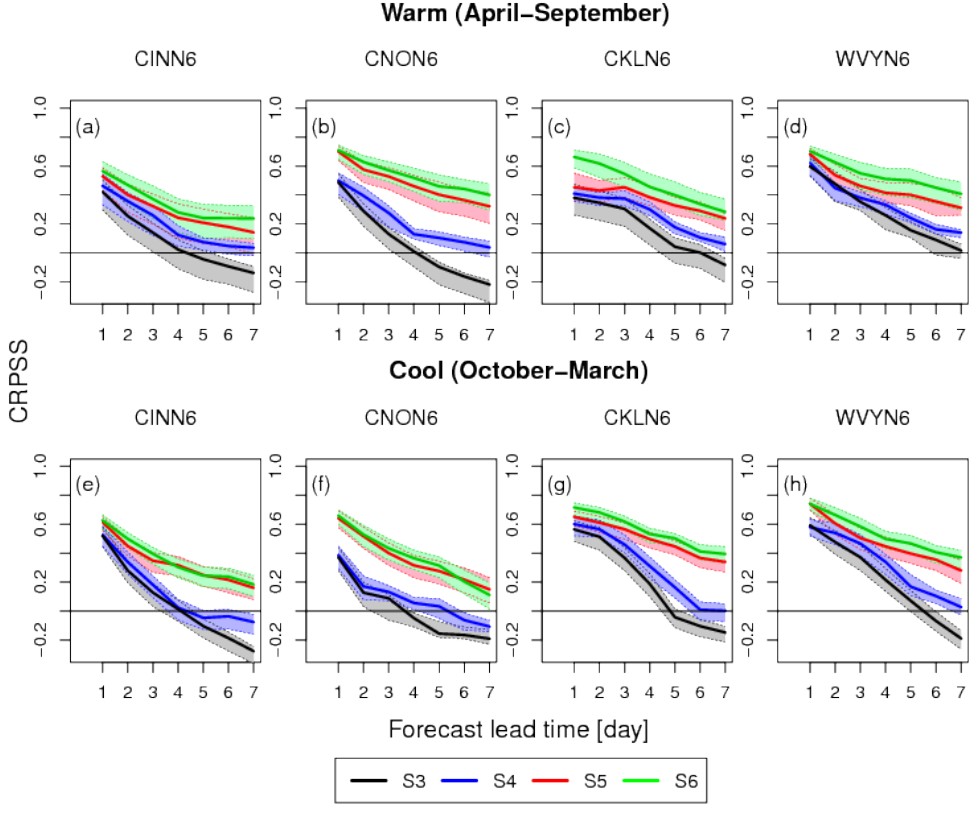

**Figure 6: As in Fig. 5, but for the CRPSS (relative to sampled climatology) of the ensemble flood forecasts vs the forecast lead time.**

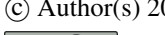


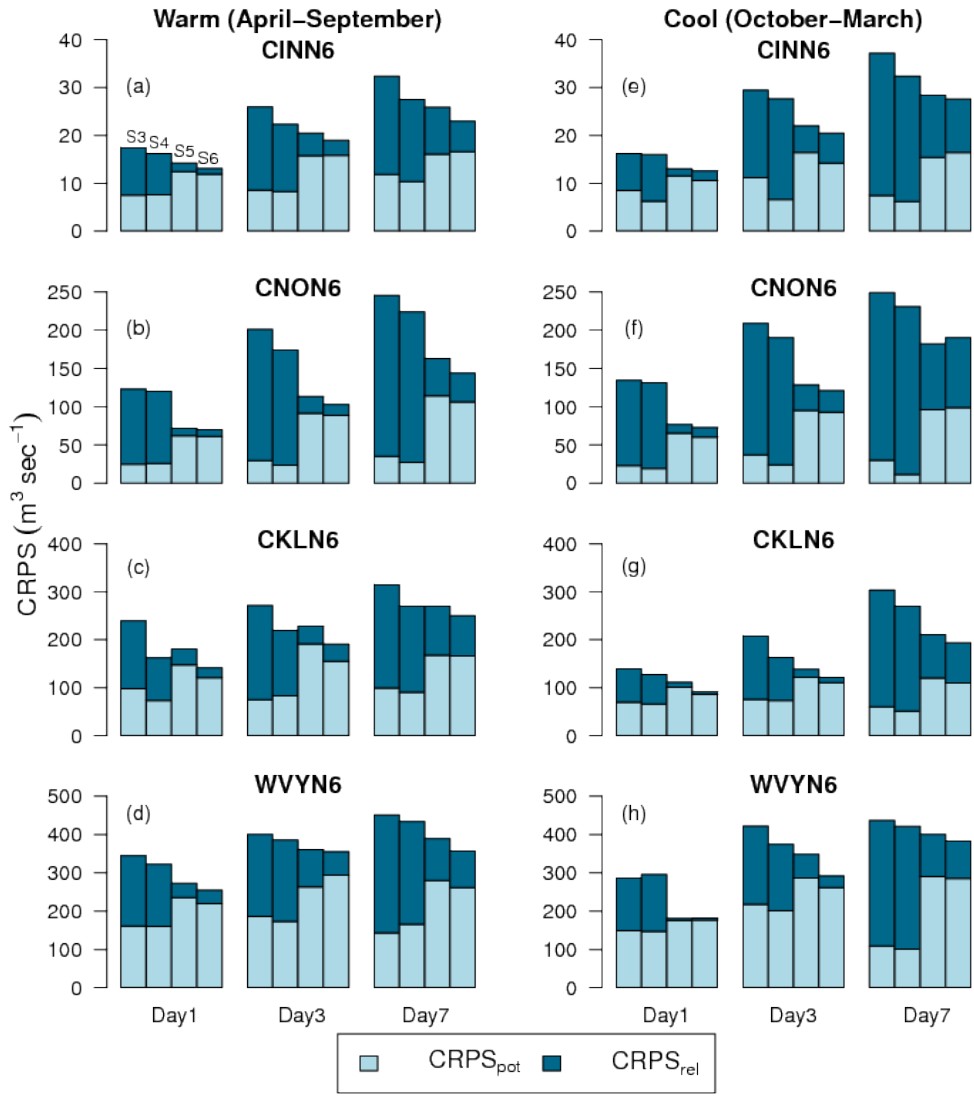

**Figure 7: Decomposition of the CRPS into CRPS potential (CRPS$_{pot}$) and CRPS reliability (CRPS$_{rel}$) for forecasts lead times of 1, 3, and 7 days during the warm (a)-(d) (April-September) and cool season (e)-(h) (October-March) for the selected basins. The four columns associated with each forecast lead time represent the forecasting scenarios S3-S6 (from left to right). Note that S3 consists of GEFSRv2+HL-RDHM, S4 of GEFSRv2+HCLR+HL-RDHM, S5 of GEFSRv2+HL-RDHM+QR, and S6 of GEFSRv2+HCLR+HL-RDHM+QR.**





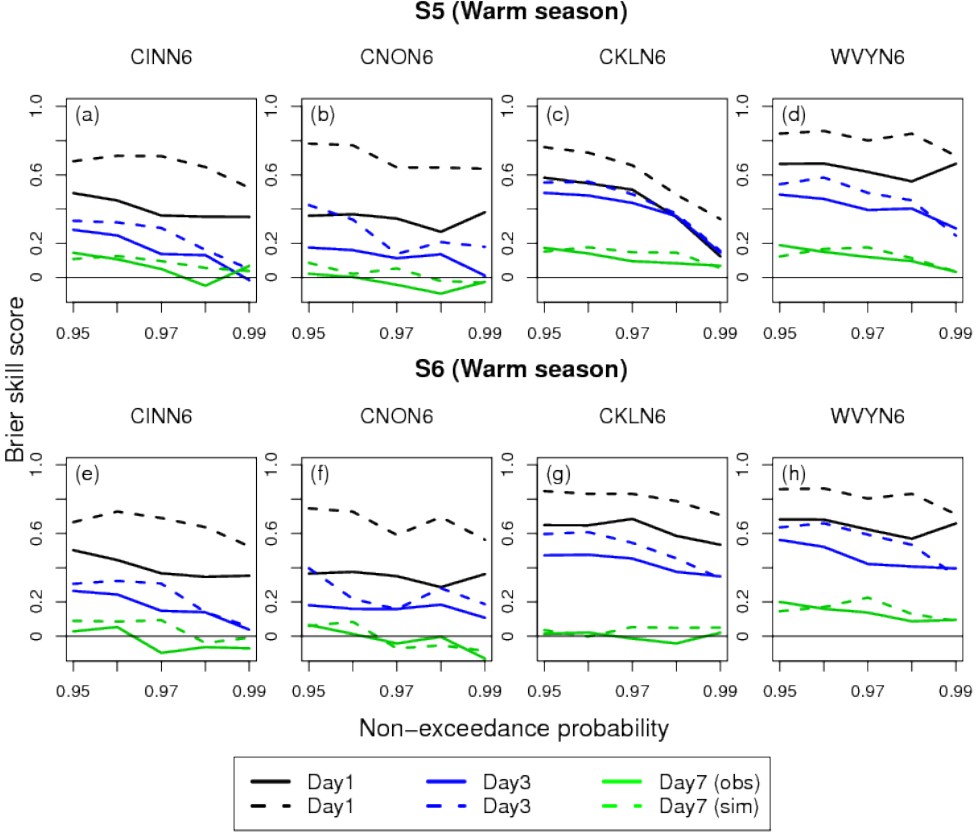

**Figure 8: Brier skill score (BSS) of the mean ensemble flood forecasts for S5 (a-d) and S6 (e-h) vs the flood threshold for forecast lead times of 1, 3, and 7 days during the warm (April-September) season for the selected basins. The BSS is shown relative to both observed (solid lines) and simulated floods (dashed lines).**

