# Peer review of "Relative effects of statistical preprocessing and postprocessing on a regional hydrological ensemble prediction system"

_Hydrology and Earth System Sciences, 2017_

## Referee Comment (RC1) · M. Scheuerer (Referee) · 13 Sep 2017

This manuscript studies the relative roles of statistical preprocessing of meteorological inputs in a hydrological forecast system and statistical postprocessing of the resulting flow forecasts for four basins in the US middle Atlantic region. The paper is well written, the structure is good, and the conclusions are interesting and relevant. The methodology is sound with two exceptions detailed below. These are major in the sense that they are scientifically problematic and may have an impact on the conclusions, but they can probably be addressed quite easily.

Specific comments:

[Figure]

- p6, l4: pi_i is only a probability when y_i=0, otherwise a likelihood

- p7, l15: 'smallest mean CRPS is selected': I don't fully understand how this works. Apparently c_{i+1} changes over time, so what exactly is minimized here? The CRPS over some training data with a rolling training window? Please add some more explanation

- p8, l15-16: '... is focused on flood events ... by choosing flow amounts greater than ...': This kind of subsetting is very problematic and can lead to false conclusions about the relative predictive performance of different methods, see Lerch et al. (2017). Bellier et al (2017) give a discussion of pitfalls of sample stratification and make suggestions how one can stratify samples in a way that avoids these pitfalls

- Section 4.4.1: I'm not sure if this part of the analysis makes sense. In addition to the stratification issue (which demonstrably entails a bias), it is also known that the ensemble mean does not necessarily yield the best/appropriate point forecast when a relative error statistic is considered (see Gneiting 2011). I suggest either considering the mean error (over the entire verification data), or omitting this subsection entirely and maybe replace it by a subsection that studies reliability of threshold exceedance

Language and typos:

- p6, l31: hourly

- p7, eq (7): xi_{I+1} -> xi_{i+1}

- p9, l15: It sounds weird to say that one basin outperforms the other, please reformulate

- p10, l24: Replace 'While' by 'The gains ... , on the other hand, ...'

References:

Bellier et al. (2017): Sample Stratification in Verification of Ensemble Forecasts of Continuous Scalar Variables: Potential Benefits and Pitfalls. Monthly Weather Review,

https://doi.org/10.1175/MWR-D-16-0487.1

Gneiting (2011): Making and Evaluating Point Forecasts. Journal of the American Statistical Association, https://doi.org/10.1198/jasa.2011.r10138

Lerch et al. (2017): Forecasters dilemma: Extreme events and forecast evaluation. Statistical Science, https://doi.org/10.1214/16-STS588.

---

## Referee Comment (RC2) · Anonymous Referee #2 · 29 Sep 2017

This review is for manuscript HESS-2017-514: Relative effects of statistical preprocessing and postprocessing on a regional hydrological ensemble prediction system, authored by Sanjib Sharma, Ridwan Siddique, Seann Reed, Peter Ahnert, Pablo Mendoza, and Alfonso Mejia. The manuscript is easy to follow. It presents some interesting results. In this work, a spatially distributed hydrological model is included in the study. Two postprocessors: an autoregressive model with a single exogenous variable and quantile regression, are comparatively evaluated. Below are my general and specific comments.

General Comments:

I was intrigued by reading the statement "postprocessing alone performs similar, in terms of the relative mean error, skill, and reliability, to the more involved scenario that includes both preprocessing and postprocessing" in the Abstract (page 1, lines 24-25). This is one of the major conclusions of the work. However, further reading reveals that the results do not fully support this conclusive statement, for the following reasons: (1) Figures 5 and 6 show appreciable performance gains of S6 over S5 for 5 cases out of 8. One can see that S6 outperforms S5 in terms of forecast lead times by 12 hours to 3 days. (2) The closeness of the results for the other cases (i.e., (e) and (f)) between S5 and S6 can be explained by the closeness of the raw GEFS and preprocessed GEFS precipitation, as shown in Figure 3. (3) The verification appears to be only conducted for large observed events without considering large forecast events, which can generate false-alarms. In short, I find this conclusion is inaccurate and can be misleading.

How are the GEFS precipitation and temperature downscaled to force the HR-RDHM? A description should be provided.

Specific Comments:

Page 1, Line 12: Do you mean "Is comprised of " by "is comprised by"?

Page 1, Line 28: In "The intersection of climate variability and change, increased exposure from expanding urbanization, and sea level rise are increasing", what do you mean by "The intersection of climate variability and change"?

Page 2, Line 6: In "for research purposes, meet specific regional needs, and/or real-time forecasting applications", do you mean "to meet . . ."?

Page 3, line 21: Shouldn't it be U.S. Middle Atlantic region?

Page 5, line 16: "Also, HCLR has been shown to outperform other widely used pre-processors (Yang et al., 2017)". Should be more specific here since the paper only compares the HCLR and BMA.

Page 6, line 31: "6-houlry" is a typo.

Page 7, line 23: "QR has similar skill performance in streamflow and normal space". This sentence is not clear to me. Do you mean that QR has similar skill performance in the streamflow space as well as normal space?

Page 8, line 15: How many events result from this threshold? Is the sampled climatological probability distribution derived from the observed data? If so, will your conclusions still hold if events corresponding to forecasts with large magnitudes and high probabilities also included in the verification?

Page 10, line 33: "QR displays better reliability than ARX(1,1) across lead times, basins, and seasons". By what measure(s)?

Page 11, line 36: "reinforcing the fact that preprocessing may have little effect on the flood forecasts". See the General Comments.

---

## Referee Comment (RC3) · Anonymous Referee #3 · 3 Oct 2017

The manuscript describes a comparative analysis of pre and post processing approaches and their contributions to flood forecasting performance in the Middle Atlantic Region. The analysis starts by evaluating the hydrology model performance. Then authors evaluate one pre processor and confirm that it improves the skill of raw precipitation forecasts. Next they evaluate two post processors and select the most performing one. Finally, authors evaluate multiple cases : raw, with or without pre and post processors. The analysis focuses on two periods for the evaluations, and 4 basins of different sizes. Authors conclude that post processing for flood forecasting is necessary and provides the largest skill increase. Pre –processing appears unnecessary.

[Figure]

The paper is very well written and organized. The approach, application and conclusion are of interest to the HESS community which has published extensively on ensemble flow forecasting. I have some moderate and minor comments below that would need to be addressed.

Moderate comments:

- The pre-processor is evaluated for 6 hourly 95th percentile events but is not evaluated for aggregated period events, which ultimately drive to floods. There is therefore a disconnection between the "value" of the post processor when evaluated independently, and the "value" of the pre –processor when verifying floods. The pre-processor has not been evaluated for the same "events".

- The conclusion that post processing only is needed to improve the skill of flow forecast seems to be based on statistics only and therefore you might get the right answer for the wrong reasons. The post processor maybe have the largest "value" but it does not mean that pre-processing steps should be skipped. I strongly recommend the authors to modify the conclusion to reflect that nuance.

- Literature review and contribution of the paper and conclusion: A HEPEX blog by Boucher A. M. (2015) provides a summary of the contribution of previous papers. She refers to the papers also mentioned below. 1) The literature and the insight provided by this experiment should be put in perspective with what has been done and found by others before.

2) The fact that spatially disaggregated modeling is used might not be enough because there is no insight related to that modeling structure to the results. I would suggest framing the contribution differently.

Boucher A. M. (2015) Pre-, post-processing or both? HEPEX Blog - (https://hepex.irstea.fr/pre-post-processing-or-both/).

Kang T.-H., Kim Y.-O. and Hong I.-P. (2010) Comparison of pre- and post-processors

for ensemble streamflow prediction, Atmospheric Science Letters, 11, 153-159.

Roulin E. and Vannitsem S. (2015) Post-processing of medium-range probabilistic hydrological forecasting: impact of forcing, initial conditions and model errors, Hydrological Processes, 29, 1434-1449.

Verkade J.S., Brown J.D., Reggiani P. and Weerts A.H. (2013) Post-processing ECMWF precipitation and temperature ensemble reforecasts for operational hydrologic forecasting at various spatial scales, Journal of Hydrology, 501, 73-91.

Zalachori I., Ramos M.-H., Garçon R., Mathevet T. and Gailhard J. (2012) Statistical processing of forecasts for hydrological ensemble prediction: a comparative study of different bias correction strategies, Advances in Science and Research, 8, 135-141.

Minor comments:

- Study domain – this corresponds to the Susquehanna Basin – why use MAR instead of the Susquehanna River Basin?

- Warm and cold seasons: can you describe the type of events expected in both seasons?

- PG 6 L31: change to "hourly"

- PG9 L4: add "observed" to "gridded precipitation"

- PG9 L4: please specify the source of the gridded observed precipitation

- PG9 L24: confusing; you mean " high precipitation events defined as 6-hourly accumulated precipitation events with a .95 non exceedance probability"? Also – see comment for the need to evaluate aggregated events

- PG10 – Line 35: how do you specify flood events? Are those also 6 hourly discharge event with a .95th non exceedance probability? Please clarify.

- Basins are not independent, could you add one comment how this might affect the

results? In the result section at PG11 L34 it looks like you could see consistent results. It did not seem to be the case on the previous section.

---

## Author Comment (AC2) · 18 Nov 2017

**Relative effects of statistical preprocessing and postprocessing on a regional hydrological ensemble prediction system**

Sanjib Sharma, Ridwan Siddique, Seann Reed, Peter Ahnert, Pablo Mendoza, Alfonso Mejia

**Response to the reviewers' comments**

We thank you very much for your thorough review of manuscript hess-2017-514. Below we provide a point-by-point response to each of the comments. The reviewers' comments are shown in blue font and our response follows immediately after that.
* * *
**Comment from Reviewer #2**: 1) I was intrigued by reading the statement "postprocessing alone performs similar, in terms of the relative mean error, skill, and reliability, to the more involved scenario that includes both preprocessing and postprocessing" in the Abstract (page 1, lines 24-25). This is one of the major conclusions of the work. However, further reading reveals that the results do not fully support this conclusive statement, for the following reasons:

i) Figures 5 and 6 show appreciable performance gains of S6 over S5 for 5 cases out of 8. One can see that S6 outperforms S5 in terms of forecast lead times by 12 hours to 3 days.
ii) The closeness of the results for the other cases (i.e., (e) and (f)) between S5 and S6 can be explained by the closeness of the raw GEFS and preprocessed GEFS precipitation, as shown in Figure 3.
iii) The verification appears to be only conducted for large observed events without considering large forecast events, which can generate false-alarms. In short, I find this conclusion is inaccurate and can be misleading.

**Response to reviewer #2**: We agree with the reviewer. We have now modified the revised manuscript to indicate that the scenario involving both preprocessing and postprocessing, i.e. S6, consistently outperforms the other scenarios. However, we also indicate that in some cases the differences between S5 (only postprocessing) and S6 are not as significant. We believe, as the reviewer suggested, that this statement and conclusion is more consistent with the overall results that are presented in the manuscript.

In regards to the reviewer's point iii), we have now revised Figures 3-7 in the new version of the manuscript by computing all the verification metrics over the entire verification period (please also see our response to reviewer 1 regarding this issue). The revised figures show more clearly that S6 is consistently better than the other scenario. Qualitatively, the revised and original figures are overall similar. But some difference do emerge, as indicated in our response to reviewer 1, particularly between the warm and cool season. We have now revised the manuscript to note and discuss these differences.

2) How are the GEFS precipitation and temperature downscaled to force the HR-RDHM? A description should be provided.
**Response to reviewer #2**: As suggested by the reviewer, we added the following text in the revised manuscript: "The GEFSRv2 data are bilinearly interpolated onto the 4 x 4 km$^2$ grid cell resolution of the HL-RDHM model."

3) Page 1, Line 12: Do you mean "Is comprised of "by "is comprised by"?
**Response to reviewer #2**: This modification was incorporated into the revised manuscript.

4) Page 1, Line 28: In "The intersection of climate variability and change, increased exposure from expanding urbanization, and sea level rise are increasing", what do you mean by "The intersection of climate variability and change"?

**Response to reviewer #2**: We meant by this statement that climate variability and climate change, which act together, alongside expanding urbanization and sea level rise are making flood prediction more challenging. We now revised the manuscript to say only "climate change" as we think this makes the sentence more clear and easier to read.

5) Page 2, Line 6: In "for research purposes, meet specific regional needs, and/or real-time forecasting applications", do you mean "to meet . . ."?

**Response to reviewer #2**: We have corrected the text following the reviewer's suggestion.

6) Page 3, line 21: Shouldn't it be U.S. Middle Atlantic region?

**Response to reviewer #2**: We incorporated this modification into the revised manuscript.

7) Page 5, line 16: "Also, HCLR has been shown to outperform other widely used preprocessors (Yang et al., 2017)". Should be more specific here since the paper only compares the HCLR and BMA.

**Response to reviewer #2**: Following the reviewer's comment, we made our statement more specific; it now reads as follows: "Also, HCLR has been shown to outperform other widely used preprocessors, such as Bayesian Model Averaging".

8) Page 6, line 31: "6-houlry" is a typo.

**Response to reviewer #2**: Thanks for catching this. We incorporated this modification into the revised manuscript.

9) Page 7, line 23: "QR has similar skill performance in streamflow and normal space". This sentence is not clear to me. Do you mean that QR has similar skill performance in the streamflow space as well as normal space?

**Response to reviewer #2**: We rephrased this sentence to incorporate the reviewer's comment. The revised sentence reads as follows: "QR is applied here in streamflow space, since it has been shown that, in hydrological forecasting applications, QR has similar skill performance in streamflow space or normal space".

10) Page 8, line 15: How many events result from this threshold? Is the sampled climatological probability distribution derived from the observed data? If so, will your conclusions still hold if events corresponding to forecasts with large magnitudes and high probabilities also included in the verification?

**Response to reviewer #2**: The reviewer makes a good point. As mentioned in one of our previous response, we have now modified the manuscript by computing the metrics (Figs. 3-7) over the entire verification period. Overall, our conclusions did not change based on the revised figures. As noted before, we do see now some seasonal differences (mainly, the performance of scenarios S4-S6 is more similar to each other in the warm season than it was before in the original manuscript) and the ability of S6 to outperform the other scenarios is more clear now.

11) Page 10, line 33: "QR displays better reliability than ARX (1,1) across lead times, basins, and seasons". By what measure(s)?

**Response to reviewer #2**: This sentence was slightly modified in the revised manuscript to add clarity and address the reviewer's comment. The new sentence reads: "We also computed reliability diagrams for the two postprocessors (plots not shown) and found that QR displays better reliability than ARX(1,1) across lead times, basins, and seasons." The figures are not shown simply to keep the length of the manuscript and number of figures manageable.

12) Page 11, line 36: "reinforcing the fact that preprocessing may have little effect on the flood forecasts". See the General Comments.

**Response to reviewer #2**: We agree with the reviewer on this comment and modified this sentence to read as follows: "However, the gain in skill between S3 and S4 is generally small, particularly at the short lead times, reinforcing the fact that preprocessing alone may have little effect on the flood forecasts."

---

## Author Comment (AC3) · 18 Nov 2017

**Relative effects of statistical preprocessing and postprocessing on a regional hydrological ensemble prediction system**

Sanjib Sharma, Ridwan Siddique, Seann Reed, Peter Ahnert, Pablo Mendoza, Alfonso Mejia

**Response to the reviewers' comments**

We thank you very much for your thorough review of manuscript hess-2017-514. Below we provide a point-by-point response to each of the comments. The reviewers' comments are shown in blue font and our response follows immediately after that.
* * *
**Comment from Reviewer #3:** 1) the pre-processor is evaluated for 6 hourly 95th percentile events but is not evaluated for aggregated period events, which ultimately drive to floods. There is therefore a disconnection between the "value" of the post processor when evaluated independently, and the "value" of the pre –processor when verifying floods. The pre-processor has not been evaluated for the same "events".

**Response to reviewer #3**: The reviewer makes a good point. As we indicated before in our response to reviewer #1 and #2, we now use in the revised manuscript all the verification values when computing the verification metrics in Figures 3-7, i.e., we do not use any threshold or stratified sample. This means that all the preprocessed precipitation values and all the postprocessed flow values are used to compute the verification metrics.

We also note that we use 6-hourly accumulations since this is the resolution of the GEFSRv2 data after day 4 and since this is a temporal resolution commonly used in operational forecasting in the U.S. In Fig. 3, we want simply to illustrate the performance of S1 and S2 relative to each other, for this purpose using 6-hourly accumulations seems reasonable (i.e., the relative comparison between S1 and S2 is similar for 6-houlry or daily accumulations). Further, we use the 6-hourly precipitation accumulations to force the hydrological model and generate 6-hourly flows. Since the observed flow data are mean daily, we compute the mean daily flow forecast from the 6-hourly flows. The postprocessor is applied to the mean daily values since this is the resolution of the observations. But there is no mismatch between precipitation and flood events.

2) The conclusion that post processing only is needed to improve the skill of flow forecast seems to be based on statistics only and therefore you might get the right answer for the wrong reasons. The post processor maybe have the largest "value" but it does not mean that pre-processing steps should be skipped. I strongly recommend the authors to modify the conclusion to reflect that nuance.

**Response to reviewer #3**: We agree with the reviewer. As suggested by the reviewer's comment, we have now modified the conclusion to read as follows: "The scenario involving both preprocessing and postprocessing consistently outperforms the other scenarios. In some cases, however, the differences between the scenario involving preprocessing and postprocessing, and the scenario with postprocessing alone, are not as significant, suggesting for those cases that postprocessing alone can be effective in removing systematic biases."

3) Literature review and contribution of the paper and conclusion: A HEPEX blog by Boucher A. M. (2015) provides a summary of the contribution of previous papers. She refers to the papers also

mentioned below. 1) The literature and the insight provided by this experiment should be put in perspective with what has been done and found by others before.

**Response to reviewer #3**: Thanks for pointing us to this blog. We were indeed aware of the blog by Boucher A. M. (2015) (https://hepex.irstea.fr/pre-post-processing-or-both/), which summarizes different papers (e.g., Kang et al. (2010), Zalachori et al. (2012), Verkade et al. (2013), and Roulin and Vannitsem (2015)) related to preprocessing and postprocessing in streamflow forecasting. In fact, we have already discussed these paper/studies and their major findings in the original manuscript. Furthermore, our research questions and experimental set-up for the manuscript were designed in part to address concerns raised in the blog.

4) The fact that spatially disaggregated modeling is used might not be enough because there is no insight related to that modeling structure to the results. I would suggest framing the contribution differently.

**Response to reviewer #3**: We agree with the reviewer. However, we do not frame the contribution in terms of going from lumped to distributed hydrological modeling. This was not our intention and it is not what we say in the original manuscript. However, we do note in the original manuscript that this is clearly one aspect of the present study that differs from previous one. It was indeed surprising to us that most previous pre/postprocessing studies have been done with lumped models. Beyond the issue of model structure indicated by the reviewer, we think it is important to mention this aspect of the study because computationally the problem becomes very different when a distributed model is used. Further, the application of the preprocessor is also very different, hence worth mentioning in our opinion that a distributed model is used.

5) Study domain – this corresponds to the Susquehanna Basin – why use MAR instead of the Susquehanna River Basin?

**Response to reviewer #3**: We agree with the reviewer and have now incorporated this modification into the revised manuscript.

6) Warm and cold seasons: can you describe the type of events expected in both seasons?

**Response to reviewer #3**: To address the reviewer's comment, we added the following information to the revised manuscript: "The climate in the upper Mid-Atlantic Region can be classified as warm, humid summers and snowy, cold winters with frozen precipitation. During the cool season, a positive North Atlantic Oscillation phase generally results in increased precipitation amounts and occurrence of heavy snow. Thus, flooding in the cool season is dominated by heavy precipitation events accompanied by snowmelt runoff. While in the summer season, convective thunderstorms with increased intensity may lead to greater variability in streamflow."

7) PG 6 L31: change to "hourly"

**Response to reviewer #3**: Thanks for catching this. We incorporated this modification into the revised manuscript.

8) PG9 L4: add "observed" to "gridded precipitation"

**Response to reviewer #3**: We incorporated this modification into the revised manuscript.

9) PG9 L4: please specify the source of the gridded observed precipitation

**Response to reviewer #3**: The information requested by the reviewer is already included in the original manuscript. The text in the original manuscript reads: "Both the MPEs and gridded near-surface air temperature data at 4 x 4 km$^2$ resolution were provided by the NOAA's Middle Atlantic River Forecast Center (MARFC)".

10) PG9 L24: confusing; you mean "high precipitation events defined as 6-hourly accumulated precipitation events with a .95 non exceedance probability"? Also – see comment for the need to evaluate aggregated events

**Response to reviewer #3**: We have now modified the original manuscript to reflect the fact that we no longer use the 0.95 threshold but instead use all the verification data. We believe this change made the sentence more clear.

11) PG10 – Line 35: how do you specify flood events? Are those also 6 hourly discharge event with a .95th non exceedance probability? Please clarify

**Response to reviewer #3**: We believe that our previous answer to the reviewer helps to address this question as well.

12) Basins are not independent, could you add one comment how this might affect the results? In the result section at PG11 L34 it looks like you could see consistent results. It did not seem to be the case on the previous section.

**Response to reviewer #3**: We believe the results will be similar if we had selected basins that are geographically close to each other and of similar size to the ones we selected. In fact, we initially selected nested sub-basins in order to investigate the forecast performance with respect to basin size or, in other words, the scaling of verification metrics with basin size. However, we found that, although there is some tendency for the larger basins to show better forecast skill than the small ones, the scaling is rather mild and not consistent. The scaling tends to show significant variability so that it is not necessarily evident for the conditions considered (e.g., lead times and seasons). This information is now mentioned in the revised manuscript.

---

## Author Response (AR1)

**Relative effects of statistical preprocessing and postprocessing on a regional hydrological ensemble prediction system**

Sanjib Sharma, Ridwan Siddique, Seann Reed, Peter Ahnert, Pablo Mendoza, Alfonso Mejia

**Response to the reviewers' comments**

We are thankful to the Editor, Dr. Shraddhanand Shukla, and the reviewers for their thorough review of manuscript hess-2017-514. We have considered each comment and suggestion made by the reviewers when revising our manuscript.  Below we provide a point-by-point response to each of the comments. The reviewers' comments are shown in blue font and our response follows immediately after that.
* * *
**RESPONSE TO EDITOR'S COMMENTS**

**Comment from the editor:** 1) Please do not exclude the summary/general statements (the first paragraph) made by the reviewers from your response. Please submit a revised version of your response that includes summary statements made by each of the reviewers.
**Response to the editor**: We have now added the summary statement made by each reviewer in the "Response to Reviewer" section.

**Comment from the editor:** 2) Reviewer #2, Comment #4: I do not think that dropping climate variability from this statement is fair. My suggestion would be to say something to the affect that both climate variability and climate change contribute to increased exposure increased exposure from expanding urbanization, and sea                  level                  rise                  are                  increasing.
**Response to the editor**: As suggested, we have now modified the sentence in P1 L30 to read as follows: "Both climate variability and climate change, increased exposure from expanding urbanization, and sea level rise are increasing the frequency of damaging flood events and making their prediction more challenging across the globe."

**Comment from the editor:** 3) Reviewer #2, Comment #10: Please specify here which comment of the reviewer #1 is similar to this comment, which you have already responded.
**Response to the editor**: We now indicate that our response to Reviewer #2 Comment #10 is similar to our response to Reviewer #1 Comment #3.

**Comment from the editor:** 4) Reviewer #2, Comment #11: It is not clear to me how the revised sentence add any clarity. My guess is that the reviewer would like you to be more specific about the metric score use for making this statement and perhaps also mention if the reliability improves for certain category of events.
**Response to the editor**:  We used the reliability diagram to quantify the reliability of the forecasts. The reliability diagram shows the full joint distribution of forecasts and observations to reveal the reliability of the probability forecasts.  Response to Comment #11 from Reviewer #2 now reads as follows: "We also computed reliability diagrams, as determined by Sharma et al. (2017),  for the two postprocessors (plots not shown) and found that QR displays better reliability than ARX(1,1) across lead times, basins, and seasons." This information is incorporated in P11 L3-5 of the revised manuscript.

**Comment from the editor:** 5) Reviewer #3, Comment #1: Please provide a reference or example for your statement "since this is a temporal resolution commonly used in operational forecasting in the U.S."
**Response to the editor**: As suggested, a link to the website from Advanced Hydrologic Prediction Center is added. Response to Comment #1 from Reviewer #3 now reads as: "We also note that we use 6-hourly accumulations since this is the resolution of the GEFSRv2 data after day 4 and since this is a temporal resolution often used in operational forecasting in the U.S (http://water.weather.gov/ahps2/hydrograph.php?wfo=bgm&gage=cinn6)." This information is incorporated in P7 L3-6 of the revised manuscript.

**Comment from the editor:** 6) Reviewer #3, Comment #4: I am not satisfied with this response, my guess would be that the reviewer may not be either. Why highlight the difference in the model structure if it has no bearing on your results? Also there are of course other previous studies that have used distributed models.

**Response to the editor**: We understand the point of the reviewer and value the comment. However, we did conduct a very exhaustive literature review. We found that most, if not all, studies that use GEFS data as forcing are with lumped or semi-distributed models. The few studies that use a distributed model tend to use ECMWF forcing, not GEFS. Even though we do not emphasize the effect of model structure in the manuscript, the use of a distributed model affects quite a bit the way the forcing is used for both the streamflow simulations and forecasts. Hence, we still think that it is worth mentioning that this is point of distinction with previous studies.

**Comment from the editor:** 7) Reviewer #3, Comment #8: Please make sure to provide more details regarding the forcing. For example please briefly mention the method, sources of observations (e.g. station or satellite or both?), which other studies have used the data before and how it was validated.

**Response to the editor**: The information requested by the editor is already included in the original manuscript. The text in the original manuscript reads: "Both the MPEs and gridded near-surface air temperature data at 4 x 4 km2 resolution were provided by the NOAA's Middle Atlantic River Forecast Center (MARFC) (Siddique and Mejia 2017). Similar to the NCEP stage-IV 5 dataset(Moore et al., 2015; Prat and Nelson, 2015), the MARFC's MPEs represent a continuous time series of hourly, gridded precipitation observations at 4 x 4 $km^2$ cells, which are produced by combining multiple radar estimates and rain gauge measurements. The gridded near-surface air temperature data at 4 x 4 km2 resolution were developed by combining multiple temperature observation networks as described by Siddique and Mejia (2017)." This information can be found in P4 L8-13 of the revised manuscript.

**RESPONSE TO REVIEWER #1**

**Comment from Reviewer #1**: This manuscript studies the relative roles of statistical preprocessing of meteorological inputs in a hydrological forecast system and statistical postprocessing of the resulting flow forecasts for four basins in the US middle Atlantic region. The paper is well written, the structure is good, and the conclusions are interesting and relevant. The methodology is sound with two exceptions detailed below. These are major in the sense that they are scientifically problematic and may have an impact on the conclusions, but they can probably be addressed quite easily.

**Response to reviewer #1**: We thank the reviewer for reviewing the manuscript. We have now addressed the reviewer's concern as detailed in the next comments.

**Comment from Reviewer #1**: 1) p6, l4: pi_i is only a probability when y_i=0, otherwise a likelihood.

**Response to reviewer #1**: We agree with the reviewer and have accordingly changed the text in the revised manuscript to read as follows: "For this, the predicted probability or likelihood $\pi_i$ of the $i^{th}$ observed outcome is determined as..." The suggested change can be found in P6 L12 of the revised manuscript.

**Comment from Reviewer #1**: 2) P7, l15: 'smallest mean CRPS is selected': I don't fully understand how this works. Apparently c_{i+1} changes over time, so what exactly is minimized here? The CRPS over some training data with a rolling training window? Please add some more explanation.

**Response to reviewer #1**: The postprocessor is implemented following a leave-one-out approach, which consists of using 7 years for training (i.e., to estimate $c_{i+1}$) and the 2 remaining years for verification purposes. This is done separately at each lead time until the entire 9 years have been verified independently from the training period. Thus, we determine a different value of $c_{i+1}$ for each 7-year training period and lead time.

To select the value of $c_{i+1}$ for each 7-year training period and lead time, we first generate ten equally spaced values of $c_{i+1}$. For each value of $c_{i+1}$, the ARX(1,1) model is trained and used to generate ensemble

streamflow forecasts, which are in turn used to compute the mean continuous ranked probability score (CRPS) for the 7-year training period under consideration. Thus, the mean CRPS is computed for each value of $c_{i+1}$, and the value of $c_{i+1}$ that produces the smallest mean CRPS is then selected for use in the 2-year verification period under consideration. This is repeated until all the years (2004-2012) have been postprocessed and verified independently of the training period. To address the reviewer's comment, we have now incorporated this explanation in P7 L24-28 of the revised manuscript.

**Comment from Reviewer #1**: 3) p8, l15-16: '... is focused on flood events ... by choosing flow amounts greater than ...': This kind of subsetting is very problematic and can lead to false conclusions about the relative predictive performance of different methods, see Lerch et al. (2017). Bellier et al (2017) give a discussion of pitfalls of sample stratification and make suggestions how one can stratify samples in a way that avoids these pitfalls.

**Response to reviewer #1**: We are thankful to the reviewer for this constructive comment. We have read the suggested papers and decided to use the entire flow values, as opposed to using a sample stratification approach, when computing the different verification metrics, with the exception of the Brier skill score. Accordingly, we revised Figures 3-7 in the new version of the manuscript. The revised figures are qualitatively similar to the previous ones. However, the revised figures are more consistent in showing the scenario involving both preprocessing and postprocessing, S6, as having better performance than the other scenarios. In addition, there are now clear differences between the warm and cool season, where the warm season shows the different scenarios, particularly S4-S6, as being more similar to each other, while the cool season results remained similar to the ones in the original manuscript. We have now modified the original manuscript in several locations to reflect the differences associated with the revised figures.

**Comment from Reviewer #1**: 4) Section 4.4.1: I'm not sure if this part of the analysis makes sense. In addition to the stratification issue (which demonstrably entails a bias), it is also known that the ensemble mean does not necessarily yield the best/appropriate point forecast when a relative error statistic is considered (see Gneiting 2011). I suggest either considering the mean error (over the entire verification data), or omitting this subsection entirely and maybe replace it by a subsection that studies reliability of threshold exceedance.

**Response to reviewer #1**: We again thank the reviewer for this constructive comment. As suggested by the reviewer, we have now removed the relative mean error statistic and this sub-section from the revised manuscript.

**Comment from Reviewer #1**: 5) P6, l31: hourly

**Response to reviewer #1**: Thanks for catching this. The typo has been corrected in the revised manuscript (see P7 L3).

**Comment from Reviewer #1**: 6) P7, eq (7): xi_{I+1} -> xi_{i+1}

**Response to reviewer #1**: Thanks for catching this. We incorporated this modification in P7 Eq. (7) of the revised manuscript.

**Comment from Reviewer #1**: 7) P9, l15: It sounds weird to say that one basin outperforms the other, please reformulate

**Response to reviewer #1**: We have now revised the text following the reviewer's suggestion. The revised sentence reads as follows (P9 L23-24): "Further, the performance of the calibrated simulation runs is similar across the four selected basins, although the largest size basin, WVYN6, shows slightly higher performance with Rm, NSE, and PB values of 0.85, 0.82, and -3%, respectively."

**Comment from Reviewer #1**: 8) p10, l24: Replace 'While' by 'The gains ..., on the other hand,'

**Response to reviewer #1**: Following the reviewer's suggestion, we incorporated this modification in P10 L36 of the revised manuscript.

**RESPONSE TO REVIEWER #2**

**Comment from Reviewer #2**: This review is for manuscript HESS-2017-514: Relative effects of statistical preprocessing and postprocessing on a regional hydrological ensemble prediction system, authored by Sanjib Sharma, Ridwan Siddique, Seann Reed, Peter Ahnert, Pablo Mendoza, and Alfonso Mejia. The manuscript is easy to follow. It presents some interesting results. In this work, a spatially distributed hydrological model is included in the study. Two postprocessors: an autoregressive model with a single exogenous variable and quantile regression, are comparatively evaluated. Below are my general and specific comments.

**Response to reviewer #2**: Thanks for reviewing the manuscript.

**Comment from Reviewer #2**: 1) I was intrigued by reading the statement "postprocessing alone performs similar, in terms of the relative mean error, skill, and reliability, to the more involved scenario that includes both preprocessing and postprocessing" in the Abstract (page 1, lines 24-25). This is one of the major conclusions of the work. However, further reading reveals that the results do not fully support this conclusive statement, for the following reasons:
i) Figures 5 and 6 show appreciable performance gains of S6 over S5 for 5 cases out of 8. One can see that S6 outperforms S5 in terms of forecast lead times by 12 hours to 3 days.
ii) The closeness of the results for the other cases (i.e., (e) and (f)) between S5 and S6 can be explained by the closeness of the raw GEFS and preprocessed GEFS precipitation, as shown in Figure 3.
iii) The verification appears to be only conducted for large observed events without considering large forecast events, which can generate false-alarms. In short, I find this conclusion is inaccurate and can be misleading.
**Response to reviewer #2**: We agree with the reviewer. We have now modified the revised manuscript to indicate that the scenario involving both preprocessing and postprocessing, S6, consistently outperforms the other scenarios. However, we also indicate that in some cases the differences between S5 (only postprocessing) and S6 are not as significant. We believe, as the reviewer suggested, that this statement and conclusion is more consistent with the overall results that are presented in the manuscript.

In regards to the reviewer's point iii), we have now revised Figures 3-7 in the new version of the manuscript by computing all the verification metrics over the entire verification period (please also see our response to reviewer # 1 comment #3 regarding this issue). The revised figures show more clearly that S6 is consistently better than the other scenarios. Qualitatively, the revised and original figures are overall similar. But some difference do emerge, as indicated in our response to reviewer 1, particularly between the warm and cool season. We have now revised the manuscript to note and discuss these differences.

**Comment from Reviewer #2**: 2) How are the GEFS precipitation and temperature downscaled to force the HR-RDHM? A description should be provided.
**Response to reviewer #2**: As suggested by the reviewer, we added the following text in the revised manuscript (P4 L25): "The GEFSRv2 data are bilinearly interpolated onto the 4 x 4 km$^2$ grid cell resolution of the HL-RDHM model."

**Comment from Reviewer #2**: 3) Page 1, Line 12: Do you mean "Is comprised of "by "is comprised by"?
**Response to reviewer #2**: This modification was incorporated into the revised manuscript (see P1 L12).

**Comment from Reviewer #2**: 4) Page 1, Line 28: In "The intersection of climate variability and change, increased exposure from expanding urbanization, and sea level rise are increasing", what do you mean by "The intersection of climate variability and change"?
**Response to reviewer #2**: We meant by this statement that climate variability and climate change, which act together, alongside expanding urbanization and sea level rise are making flood prediction more challenging. We now revised the manuscript (see P1 L30) to say "Both climate variability and climate change" as we think this makes the sentence clearer and easier to read.

**Comment from Reviewer #2**: 5) Page 2, Line 6: In "for research purposes, meet specific regional needs, and/or real-time forecasting applications", do you mean "to meet . . ."?

**Response to reviewer #2**: In P2 L9 of the revised manuscript, the suggested change was incorporated.

**Comment from Reviewer #2**: 6) Page 3, line 21: Shouldn't it be U.S. Middle Atlantic region?

**Response to reviewer #2**: We incorporated this modification into the revised manuscript (P3 L25).

**Comment from Reviewer #2**: 7) Page 5, line 16: "Also, HCLR has been shown to outperform other widely used preprocessors (Yang et al., 2017)". Should be more specific here since the paper only compares the HCLR and BMA.

**Response to reviewer #2**: Following the reviewer's comment, we made our statement more specific; it now reads as follows (P5 L25): "Also, HCLR has been shown to outperform other widely used preprocessors, such as Bayesian Model Averaging".

**Comment from Reviewer #2**: 8) Page 6, line 31: "6-houlry" is a typo.

**Response to reviewer #2**: Thanks for catching this. We incorporated this modification into the revised manuscript (P7 L3).

**Comment from Reviewer #2**: 9) Page 7, line 23: "QR has similar skill performance in streamflow and normal space". This sentence is not clear to me. Do you mean that QR has similar skill performance in the streamflow space as well as normal space?

**Response to reviewer #2**: We rephrased this sentence to incorporate the reviewer's comment. The revised sentence reads as follows (P8 L1-2): "QR is applied here in streamflow space, since it has been shown that, in hydrological forecasting applications, QR has similar skill performance in streamflow space as well as normal space (López et al., 2014)."

**Comment from Reviewer #2**: 10) Page 8, line 15: How many events result from this threshold? Is the sampled climatological probability distribution derived from the observed data? If so, will your conclusions still hold if events corresponding to forecasts with large magnitudes and high probabilities also included in the verification?

**Response to reviewer #2**: The reviewer makes a good point. We have now modified the manuscript by computing the metrics (Figs. 3-7) over the entire verification period. Overall, our conclusions did not change based on the revised figures. As noted before, we do see now some seasonal differences (mainly, the performance of scenarios S4-S6 is more similar to each other in the warm season than it was before in the original manuscript) and the ability of S6 to outperform the other scenarios is more clear now. Below we show in italic our complete answer to comment # 3 from Reviewer #1 which we think applies here as well.

*We have decided to use the entire flow values, as opposed to using a sample stratification approach, when computing the different verification metrics, with the exception of the Brier skill score. Accordingly, we revised Figures 3-7 in the new version of the manuscript. The revised figures are qualitatively similar to the previous ones. However, the revised figures are more consistent in showing the scenario involving both preprocessing and postprocessing (scenario 6) as having better performance than the other scenarios. In addition, there are now clear differences between the warm and cool season, where the warm season shows the different scenarios, particularly S4-S6, as being more similar to each other, while the cool season results remained similar to the ones in the original manuscript. We have now modified the original manuscript in several locations to reflect the differences associated with the revised figures.*

**Comment from Reviewer #2**: 11) Page 10, line 33: "QR displays better reliability than ARX (1,1) across lead times, basins, and seasons". By what measure(s)?

**Response to reviewer #2**: This sentence was slightly modified in the revised manuscript to add clarity and address the reviewer's comment. The new sentence reads (P11 L3-5): "We also computed reliability diagrams, as determined by Sharma et al. (2017), for the two postprocessors (plots not shown) and found that QR displays better reliability than ARX(1,1) across lead times, basins, and seasons." The figures are not shown simply to keep the length of the manuscript and number of figures manageable.

**Comment from Reviewer #2**: 12) Page 11, line 36: "reinforcing the fact that preprocessing may have little effect on the flood forecasts". See the General Comments.

**Response to reviewer #2**: Following the reviewer's comment, we removed the sentence from the revised manuscript.
* * *
**RESPONSE TO REVIEWER #3**

**Comment from Reviewer #3:** The manuscript describes a comparative analysis of pre and post processing approaches and their contributions to flood forecasting performance in the Middle Atlantic Region. The analysis starts by evaluating the hydrology model performance. Then authors evaluate one pre processor and confirm that it improves the skill of raw precipitation forecasts. Next they evaluate two post processors and select the most performing one. Finally, authors evaluate multiple cases: raw, with or without pre and post processors. The analysis focuses on two periods for the evaluations, and 4 basins of different sizes. Authors conclude that post processing for flood forecasting is necessary and provides the largest skill increase. Pre – processing appears unnecessary.
The paper is very well written and organized. The approach, application and conclusion are of interest to the HESS community which has published extensively on ensemble flow forecasting. I have some moderate and minor comments below that would need to be addressed.

**Response to reviewer #3**: We are thankful to the reviewer for taking the time to review our manuscript.

**Comment from Reviewer #3:** 1) the pre-processor is evaluated for 6 hourly 95th percentile events but is not evaluated for aggregated period events, which ultimately drive to floods. There is therefore a disconnection between the "value" of the post processor when evaluated independently, and the "value" of the pre – processor when verifying floods. The pre-processor has not been evaluated for the same "events".

**Response to reviewer #3**: The reviewer makes a good point. As we indicated before in our response to reviewer #1 and #2, we now use in the revised manuscript all the verification values when computing the verification metrics in Figures 3-7, i.e., we do not use any threshold or stratified sample. This means that all the preprocessed precipitation values and all the postprocessed flow values are used to compute the verification metrics.

We also note that we use 6-hourly accumulations since this is the resolution of the GEFSRv2 data after day 4 and since this is a temporal resolution often used in operational forecasting in the U.S. (http://water.weather.gov/ahps2/hydrograph.php?wfo=bgm&gage=cinn6 ). In Fig. 3, we want simply to illustrate the performance of S1 and S2 relative to each other, for this purpose using 6-hourly accumulations seems reasonable (i.e., the relative comparison between S1 and S2 is similar for 6-houlry or daily accumulations). Further, we use the 6-hourly precipitation accumulations to force the hydrological model and generate 6-hourly flows. Since the observed flow data are mean daily, we compute the mean daily flow forecast from the 6-hourly flows. The postprocessor is applied to the mean daily values since this is the resolution of the observations. But there is no mismatch between precipitation and flood events. This information was incorporated in P7 L3-6 of the revised manuscript.

**Comment from Reviewer #3:** 2) The conclusion that post processing only is needed to improve the skill of flow forecast seems to be based on statistics only and therefore you might get the right answer for the wrong reasons. The post processor maybe have the largest "value" but it does not mean that pre-processing steps should be skipped. I strongly recommend the authors to modify the conclusion to reflect that nuance.

**Response to reviewer #3**: We agree with the reviewer. As suggested by the reviewer's comment, we have now modified the conclusion to read as follows (P13 L28-31): "The scenario involving both preprocessing and postprocessing consistently outperforms the other scenarios. In some cases, however, the differences between the scenario involving preprocessing and postprocessing, and the scenario with postprocessing alone, are not as significant, suggesting for those cases that postprocessing alone can be effective in removing systematic biases."

**Comment from Reviewer #3:**  3) Literature review and contribution of the paper and conclusion: A HEPEX blog by Boucher A. M. (2015) provides a summary of the contribution of previous papers. She refers to the papers also mentioned below. i) The literature and the insight provided by this experiment should be put in perspective with what has been done and found by others before.

**Response to reviewer #3**: Thanks for pointing us to this blog. We were indeed aware of the blog by Boucher A. M. (2015) (https://hepex.irstea.fr/pre-post-processing-or-both/), which summarizes different papers (e.g., Kang et al. (2010), Zalachori et al. (2012), Verkade et al. (2013), and Roulin and Vannitsem (2015)) related to preprocessing and postprocessing in streamflow forecasting. In fact, we have already discussed these paper/studies and their major findings in the original manuscript. Furthermore, our research questions and experimental set-up for the manuscript were designed in part to address concerns raised in the blog.

ii) The fact that spatially disaggregated modeling is used might not be enough because there is no insight related to that modeling structure to the results. I would suggest framing the contribution differently.

**Response to reviewer #3**: We agree with the reviewer. It is not our intention to frame the contribution in terms of going from lumped to distributed hydrological modeling. However, we do note in the original manuscript that this is one aspect of the present study that differs from previous one. It was indeed surprising to us that most previous pre/postprocessing studies that use GEFS forcing have been done with lumped or semi-distributed models. Beyond the issue of model structure indicated by the reviewer, we think it is useful to mention this aspect of the study because the use of the forcing for simulating and forecasting streamflow is different than with a lumped model, and the application of the preprocessor is also different.

**Comment from Reviewer #3:** 4) Study domain – this corresponds to the Susquehanna Basin – why use MAR instead of the Susquehanna River Basin?

**Response to reviewer #3**: We agree with the reviewer and have now incorporated this modification into the revised manuscript (see P3 L25-37).

**Comment from Reviewer #3:** 5) Warm and cold seasons: can you describe the type of events expected in both seasons?

**Response to reviewer #3**: To address the reviewer's comment, we added the following information to the revised manuscript (P3 L28-32): "The climate in the upper MAR, where the NBSR basin is located, can be classified as warm, humid summers and snowy, cold winters with frozen precipitation (Polsky et al, 2000). During the cool season, a positive North Atlantic Oscillation phase generally results in increased precipitation amounts and occurrence of heavy snow (Durkee et al., 2007). Thus, flooding in the cool season is dominated by heavy precipitation events accompanied by snowmelt runoff. In the summer season, convective thunderstorms with increased intensity may lead to greater variability in streamflow."

**Comment from Reviewer #3:** 6) PG 6 L31: change to "hourly"

**Response to reviewer #3**: Thanks for catching this. We incorporated this modification into the revised manuscript (P7 L3).

**Comment from Reviewer #3:** 7) PG9 L4: add "observed" to "gridded precipitation"

**Response to reviewer #3**: We incorporated this modification in P9 L13 of the revised manuscript.

**Comment from Reviewer #3:** 8) PG9 L4: please specify the source of the gridded observed precipitation

**Response to reviewer #3**: The information requested by the reviewer is already included in the original manuscript. The text in the revised manuscript can be found in P4 L8-9 and reads as follows: "Both the MPEs and gridded near-surface air temperature data at 4 x 4 km$^2$ resolution were provided by the NOAA's Middle Atlantic River Forecast Center (MARFC) (Siddique and Mejia 2017)."

**Comment from Reviewer #3:** 9) PG9 L24: confusing; you mean "high precipitation events defined as 6-hourly accumulated precipitation events with a .95 non exceedance probability"? Also – see comment for the need to evaluate aggregated events

**Response to reviewer #3**: We have now modified the original manuscript to reflect the fact that we no longer use the 0.95 threshold but instead use all the verification data. We believe this change made the sentence more clear.

**Comment from Reviewer #3:** 10) PG10 – Line 35: how do you specify flood events? Are those also 6 hourly discharge event with a .95th non exceedance probability? Please clarify

**Response to reviewer #3**: We believe that our previous answer to the reviewer helps to address this question as well.

**Comment from Reviewer #3:** 11) Basins are not independent, could you add one comment how this might affect the results? In the result section at PG11 L34 it looks like you could see consistent results. It did not seem to be the case on the previous section.

**Response to reviewer #3**: We believe the results will be similar if we had selected basins that are geographically close to each other and of similar size to the ones we selected. In fact, we initially selected nested sub-basins in order to investigate the forecast performance with respect to basin size or, in other words, the scaling of verification metrics with basin size. However, we found that, although there is some tendency for the larger basins to show better forecast skill than the small ones, the scaling is rather mild and not consistent. The scaling tends to show significant variability so that it is not necessarily evident for the conditions considered (e.g., lead times and seasons). This information is now mentioned in the revised manuscript to read as follows (P11 L16-18): "Although there is some tendency for the large basins to show better forecast skill than the small ones, the scaling (i.e., the dependence of skill on the basin size) is rather mild and not consistent across the four basins.

**List of major changes made in the manuscript (hess-2017-514)**

The major changes that were incorporated into the revised manuscript are as follows:

- ➢ We now use the entire flow values, as opposed to using a sample stratification approach, when computing the different verification metrics, with the exception of the Brier skill score. Accordingly, we revised Figures 3-7 in the revised manuscript.
- ➢ We made changes throughout the result section of the manuscript to reflect the results shown in the revised Figures 3-7. The main difference in the revised figures with respect to the original ones is that seasonal differences become more obvious between the warm and cool season verification results. In addition, the verification results for the difference scenarios are more clear now than before.
- ➢ We removed from the manuscript the relative mean error statistic and the corresponding subsection since this was suggested by one of the reviewers.
- ➢ The conclusions of the paper were adjusted to reflect the result changes associated with using the entire flow values for the verification analysis.

[revised manuscript text omitted]

The underperformance of S4 in the CNON6 basin (Fig. 5f), relative to the other scenarios, is in part due to the unusually low skill of the raw ensemble streamflow forecasts of S3, so that even after preprocessing the skill improvement attained with S4 is not comparable to that associated with S5 and S6. This is also the case for CNON6 in the warm season (Fig. 5b). However, in addition, during the cool season it is likely that streamflows in CNON6 are affected by a reservoir just upstream from the main outlet of CNON6. The reservoir is operated for flood control purposes. The reservoir affects during the cool season low flows by maintaining them somewhat higher than in natural conditions. Since we do not account for reservoir operations in our hydrological modeling, it is likely that part of the benefits of postprocessing are in this case to correct for this modeling bias. In fact, this is also reflected in the The two most striking features of Fig. 5 are: i) the significant difference in performance between the pair S3-S4 and S5-S6 and, in contrast, ii) the similarity in performance between S5 and S6. The former confirms that statistical processing, in particular postprocessing, has a significant effect on the streamflow ensembles. Recall that to generate the ensemble streamflow

forecasts S5 only employs postprocessing, while S6 considers both preprocessing and postprocessing (Table 1). Yet, the CRPSS across basins, lead times, and seasons for both S5 and S6 are quite similar, with differences tending to be not as significant. The similarity between S5 and S6 indicates that in this case preprocessing has a mild effect on the streamflow forecasts.

Further, comparing the ensemble streamflow forecasts of S5 and S6 against each other, it appears that the general tendency is for the both scenarios to perform similarly. The S6, however, tends to show slight skill gain over S5 across all the basins and lead times, particularly in the cool season. While in the warm season, most part of the forecasting scenarios from S4 to S6 behave similarly with respect to each other. Indeed, S6 exhibit marginal skill gain at initial lead times (< day 3), but the skill performance becomes similar among S4 to S6 at later lead times, with an exception for CINN6 (Fig. 5a). In the case of CINN6 (Fig. 5a), S5 and S6 exhibit skill gain over S4 after lead times of 3 days.

In terms of the warm and cool seasons seasonal analysis, at the initial forecast lead times (≤ 2 days), the skill of the flood streamflow forecasts tends to be slightly greater in the cool warm season (Figs. 56ae dh) than in the coolwarm one (Figs. 56ea hd) across all the basins and lead times., with the exception of CNON6. The CNON6 exhibit lowest skill during both warm (Fig. 5b) and cool (Fig. 5f) seasons. This may be partly, among other potential factors, because of the effect of the Whitney Point reservoir that is located just upstream of the main outlet of the CNON6. The reservoir is mainly operated for flood control, thus modify the streamflow forecast at the basin outlet. As was the case in the calibration results (e.g., in Fig. 2c), where the performance of during the cool season CNON6 is somewhat lower than in the other basins.has a lower performance prior to postprocessing (S3 or S4 in Fig. 56f) than the other basins. Interestingly, after postprocessing (S5 in Fig. 56f), the skill of CNON6 is as good as that of CINN6, even though at the day 1 lead time the skill for S3 is ~0.13 for CNON6 (Fig. 56f) and ~0.45 for CINN6 (Fig. 56e). Hence, the postprocessor seems capable to compensate some for the lesser performance of CNON6 induring both calibration or after preprocessing in the cool season.

**4.4.23 CRPS decomposition**

[revised manuscript text omitted]

---

## Author Response (AR2)

**Relative effects of statistical preprocessing and postprocessing on a regional hydrological ensemble prediction system**

Sanjib Sharma, Ridwan Siddique, Seann Reed, Peter Ahnert, Pablo Mendoza, Alfonso Mejia

Our response to the one additional comment made by reviewer #1 is below.

**Comment from Reviewer #1**: My major concern has been addressed, but I still struggle to understand the methodology described in Sect. 3.5.1. The notation suggests that c is different for each time step, and unlike for the other variables there is no equation that relates the different time steps. So if ten equally spaced values are selected, is this done for every single time step independently? This would be seem like an extremely over-parametrized optimization problem. I'm probably still missing something, maybe the authors can further clarify this point.

**Response to reviewer #1:** The regression coefficient $c$ in the ARX(1, 1) model (see section 3.5.1) is estimated independently for each individual forecast lead time. For this, ten equally spaced values of $c$ are selected for every single lead time, and the optimal value of $c$ is generated by minimizing the mean CRPS. The ARX (1,1) model is being used operationally in the U.S. to statistically postprocess streamflow (Regonda et al., 2013). Our interest here was to compare this operational model against a similar alternative (similar in terms of computational requirements), i.e., quantile regression. In this way, we did not try to improve or modify the ARX(1,1) model but wanted to use the version proposed by (Regonda et al., 2013). However, the over-parametrization could be an important issue to explore further in the near future. Following the reviewer's comments, the manuscript was modified as follows (Page 7, Lines 29-30):

[revised manuscript text omitted]